# A selective inhibitor of ceramide synthase 1 reveals a novel role in fat metabolism

Nigel Turner[1], Xin Ying Lim[2,3], Hamish D. Toop [4], Brenna Osborne [1], Amanda E. Brandon[5], Elysha N. Taylor[4], Corrine E. Fiveash[1], Hemna Govindaraju[1], Jonathan D. Teo[3], Holly P. McEwen[3], Timothy A. Couttas[3], Stephen M. Butler[4], Abhirup Das[1], Greg M. Kowalski [6], Clinton R. Bruce[6], Kyle L. Hoehn [7], Thomas Fath[1,10], Carsten Schmitz-Peiffer[8], Gregory J. Cooney[5], Magdalene K. Montgomery[1], Jonathan C. Morris [4] & Anthony S. Don[3,9]

Specific forms of the lipid ceramide, synthesized by the ceramide synthase enzyme family, are believed to regulate metabolic physiology. Genetic mouse models have established C16 ceramide as a driver of insulin resistance in liver and adipose tissue. C18 ceramide, synthesized by ceramide synthase 1 (CerS1), is abundant in skeletal muscle and suggested to promote insulin resistance in humans. We herein describe the first isoform-specific ceramide synthase inhibitor, P053, which inhibits CerS1 with nanomolar potency. Lipidomic profiling shows that P053 is highly selective for CerS1. Daily P053 administration to mice fed a high-fat diet (HFD) increases fatty acid oxidation in skeletal muscle and impedes increases in muscle triglycerides and adiposity, but does not protect against HFD-induced insulin resistance. Our inhibitor therefore allowed us to define a role for CerS1 as an endogenous inhibitor of mitochondrial fatty acid oxidation in muscle and regulator of whole-body adiposity.

[1] School of Medical Sciences, UNSW Sydney, Sydney 2052 NSW, Australia. [2] Prince of Wales Clinical School, Faculty of Medicine, UNSW Sydney, Sydney 2052 NSW, Australia. [3] Centenary Institute, The University of Sydney, Sydney 2006 NSW, Australia. [4] School of Chemistry, UNSW Sydney, Sydney 2052 NSW, Australia. [5] Sydney Medical School, Charles Perkins Centre, University of Sydney, Sydney 2006 NSW, Australia. [6] Institute for Physical Activity and Nutrition, School of Exercise and Nutrition Sciences, Deakin University, Burwood 3125 VIC, Australia. [7] School of Biotechnology and Biomolecular Sciences, UNSW Sydney, Sydney 2052 NSW, Australia. [8] Garvan Institute of Medical Research, Sydney 2010 NSW, Australia. [9] NHMRC Clinical Trials Centre, Sydney Medical School, The University of Sydney, Sydney 2006 NSW, Australia[10] Present address: Department of Biomedical Sciences, Macquarie University, Sydney, NSW 2109, Australia. These authors contributed equally: Nigel Turner, Xin Ying Lim. Correspondence and requests for materials should be addressed to N.T. (email: n.turner@unsw.edu.au) or to J.C.M. (email: jonathan.morris@unsw.edu.au) or to A.S.D. (email: anthony.don@sydney.edu.au)

Ceramide is the central metabolite of the sphingolipid family, structurally comprised of a sphingoid base—generally 18 carbon dihydrosphingosine or sphingosine—with a variable length fatty acyl side-chain[1,2]. Ceramides form the lipid backbone to which a diverse array of headgroup structures are conjugated, forming sphingomyelin (SM), glucosyl- and galactosylceramide (HexCer), gangliosides, and globosides[2,3]. Ceramides are also signalling molecules that regulate ER stress[4], apoptosis[5], insulin sensitivity[1,6], and other physiological functions. At the molecular level, ceramides influence membrane fluidity, modulating the compartmentalisation of cellular signalling processes[7,8], and directly activate specific protein kinases and phosphatases such as the ubiquitous phosphatase PP2A[1,3,8]. Increased ceramide levels are heavily implicated in the pathogenesis of insulin resistance[9–12] and neurodegenerative conditions[13], whilst decreased levels fuel cancer cell resistance to therapy[3].

Ceramide synthesis in mammals is catalysed by a family of six ceramide synthases (CerS1-6). These enzymes transfer a variable length fatty acyl-coenzyme A (CoA) to the amine group of a sphingoid base[1]. Studies employing genetic manipulations have demonstrated that different CerS isoforms exhibit strong preference for fatty acyl-CoAs with differing carbon chain lengths. CerS1 exclusively uses 18 carbon (C18) fatty acids, forming C18 (d18:1/18:0) ceramide[14–16], whilst CerS2 preferentially forms d18:1/24:0 (C24:0) and d18:1/24:1 (C24:1) ceramides[16–18]. Thus, ceramide is not a single lipid entity; rather it is a family of signalling lipids with important physiological functions, and variation in the ceramide acyl-chain dramatically influences the biological properties of these lipids. Insulin resistance caused by a HFD is alleviated by CerS5 or CerS6 gene deletion, which prevents C16 ceramide synthesis in liver and adipose tissue[9,10,12]. C16 and shorter chain ceramides antagonise the insulin receptor—PI$_3$ kinase—Akt signalling pathway and inhibit fat utilisation as an energy source via β-oxidation[1,7,9,19]. In direct contrast, C24 ceramides synthesized by CerS2 protect against insulin resistance[6,9,20]. The synthesis of ceramides and other sphingolipids is regulated by availability of fatty acyl-CoA substrates, particularly palmitoyl-CoA derived from the common saturated fatty acid palmitate[2,21]. As such, ceramide synthesis may act as a direct metabolic sensor of fatty acid availability, feeding back to regulate metabolic processes.

Another ceramide species implicated in insulin resistance is C18:0[11,22,23]. CerS1 and its product C18 ceramide are highly abundant in skeletal muscle (SkM)[1,18]. Studies comparing obese insulin resistant and insulin sensitive subjects, exercise interventions in type 2 diabetes, and induction of insulin resistance in mice, all show an association between muscle C18:0 ceramide and impairments in insulin action[11,22,24]. Although a relatively minor species in plasma, circulating C18 ceramide is also very significantly correlated with body mass index[25] and visceral fat mass[22]. Similarly, C18 ceramide in SkM is positively correlated with visceral fat, as well as blood pressure[22].

Selective inhibition of CerS5, CerS6, and/or CerS1 would therefore be predicted to produce significant benefits for metabolic health, whilst CerS2 inhibition would have detrimental effects. However, isoform-specific CerS inhibitors with sufficient potency, selectivity, and bioavailability for in vivo use have not yet been developed[26,27]. This report describes the discovery and characterisation of the first potent, isoform-selective CerS inhibitor, specifically targeting CerS1. CerS1 inhibition is shown to promote fatty acid oxidation in SkM and reduce overall adiposity in mice fed a HFD.

## Results

**Development of a potent and selective CerS1 inhibitor.** To develop isoform-selective CerS inhibitors, we started with the multiple sclerosis drug Fingolimod (FTY720, Gilenya), which is an analogue of the endogenous lipid sphingosine[28]. Following its phosphorylation in vivo, Fingolimod is a potent agonist of sphingosine 1-phosphate receptors, however the non-phosphorylated pro-drug also exhibits non-selective inhibition of ceramide synthases as an off-target effect[29,30]. We recently established that the non-phosphorylatable, chiral FTY720 analogue AAL(S) and its benzyl tail derivative G024 (Fig. 1a, compound **1**), show limited selectivity for CerS1 over other CerS isoforms[27]. However, their degree of selectivity for CerS1 is poor and, importantly, these compounds do not selectively reduce C18 ceramide levels in cultured cells. Using the Topliss tree as a guide[31,32] we examined variations of the benzyl tail of G024, resulting in the identification of (S)-2-amino-4-(4-(3,4-dichlorobenzyloxy)phenyl)-2-methylbutan-1-ol (P053, [compound **2**]) as a potent and selective CerS1 inhibitor (Fig. 1a, b). The IC$_{50}$ for inhibition of CerS1 with P053 was 0.5 μM, an order of magnitude lower than that of G024 (Table 1). IC$_{50}$'s for inhibition of CerS2, CerS4, CerS5, and CerS6 by P053 were all at least one order of magnitude higher than for CerS1, demonstrating strong selectivity for CerS1 over other CerS isoforms (Fig. 1b and Table 1). The IC$_{50}$ for P053 on CerS1 was similar to that of the potent but non-selective CerS inhibitor Fumonisin B1 (FB1)[33], which inhibited all CerS isoforms with sub-micromolar IC$_{50}$ (Table 1). As shown previously by Lahiri et al.[29] for CerS2, CerS1 activity as

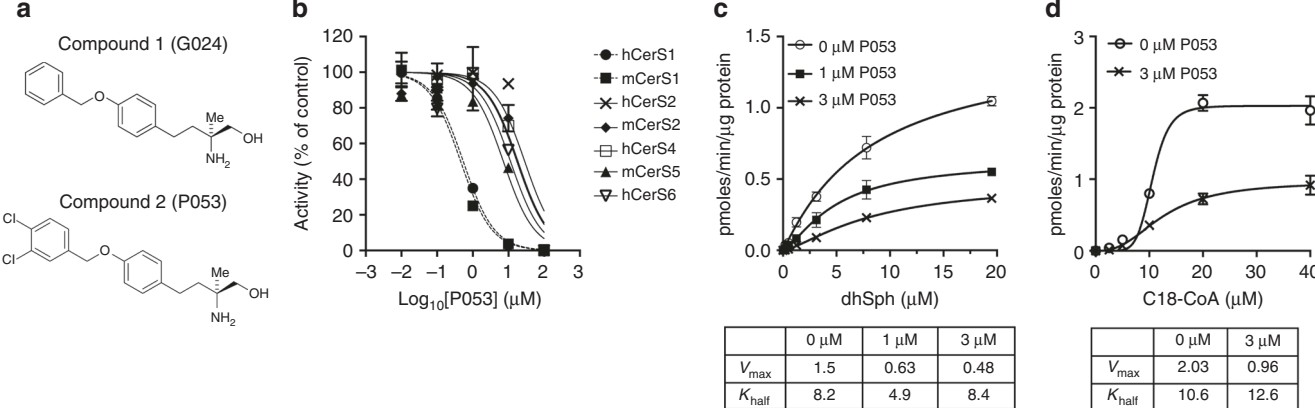

**Fig. 1** P053 is a selective inhibitor of CerS1. **a** Structure of G024 [1] and P053 [2]. **b** Activity of CerS isoforms as a function of P053 concentration (n = 3 for each concentration, mean ± SEM). **c**, **d** CerS1 activity as a function of **c** dihydrosphingosine or **d** C18:0-CoA concentration. Calculated V$_{max}$ and K$_{half}$ values are below the graphs. Graphs show mean ± SEM (n = 3 for each concentration), and the non-competitive inhibition was confirmed in three independent experiments for each substrate

| | 0 μM | 1 μM | 3 μM |
|---|---|---|---|
| V$_{max}$ | 1.5 | 0.63 | 0.48 |
| K$_{half}$ | 8.2 | 4.9 | 8.4 |

| | 0 μM | 3 μM |
|---|---|---|
| V$_{max}$ | 2.03 | 0.96 |
| K$_{half}$ | 10.6 | 12.6 |

| | hCerS1 | mCerS1 | hCerS2 | mCerS2 | hCerS4 | mCerS5 | hCerS6 |
|---|---|---|---|---|---|---|---|
| **Table 1 IC$_{50}$ for inhibition of ceramide synthases by P053** | | | | | | | |
| P053 [2] | 0.54 ± 0.06 | 0.46 ± 0.08 | 28.6 ± 0.15 | 18.5 ± 0.12 | 17.2 ± 0.09 | 7.2 ± 0.10 | 11.4 ± 0.17 |
| G024 [1] | 6.0 ± 0.10 | | 14.5 ± 0.09 | | 18.0 ± 0.12 | 24.3 ± 0.22 | 8.5 ± 0.13 |
| FTY720 | 9.9 ± 0.07 | 5.6 ± 0.05 | 19.1 ± 0.10 | 10.6 ± 0.13 | 26.0 ± 0.18 | 15.0 ± 0.07 | 5.8 ± 0.20 |
| FB1 | 0.22 ± 0.05 | | 0.15 ± 0.08 | | 0.92 ± 0.06 | 0.50 ± 0.05 | 0.29 ± 0.05 |

Calculated IC$_{50}$ values (µM) for inhibition of different human (h) or murine (m) CerS isoforms by P053, G024, FTY720, and FB1. IC$_{50}$ values were calculated using an inhibitor range from 10 nM–100 µM, as shown in Fig. 1b

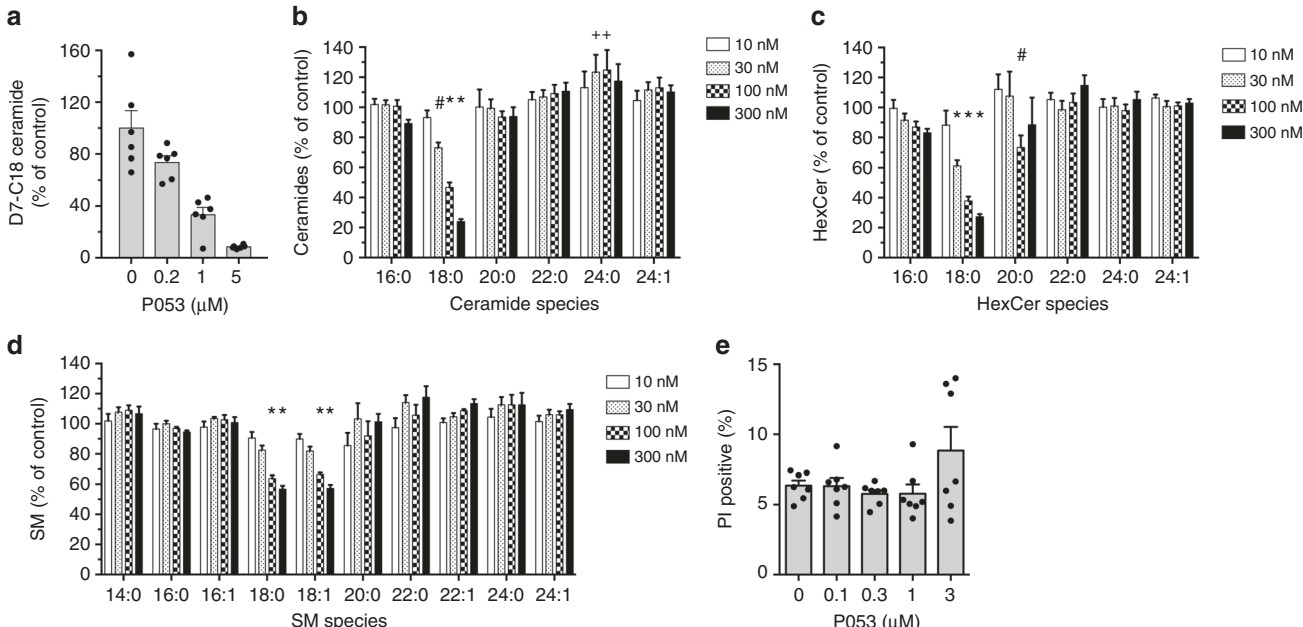

**Fig. 2** P053 selectively reduces C18 sphingolipids in cultured cells. **a** Formation of deuterated ceramide from deuterated (D7) dihydrosphingosine in cortical neuron cultures pre-treated for 2 h with P053. **b–d** Endogenous ceramide (**b**), HexCer (**c**), and SM (**d**) levels in HEK293 cells treated for 24 h with P053 at concentrations indicated on the graphs. Lipid levels are expressed as a percentage of vehicle control levels. Absolute lipid levels are shown in Supplementary Figure 1, with raw data for these images provided in Supplementary Data File 1. **e** Percentage of non-viable HEK293 cells determined by propidium iodide (PI) staining and flow cytometry, following a 72 h treatment with P053. All graphs show combined data from two independent experiments, each with 3 (**a**) or 4 (**b–d**) separate cell treatments, mean ± SEM. Statistical significance in (**b–d**) was determined by *t*-tests adjusted for multiple comparisons, comparing all lipids (22 lipids tested in total) at each concentration to the control group; +$P$ < 0.05; #$P$ < 0.01; *$P$ < 0.001

a function of substrate concentration is sigmoidal, particularly for the C18:0-CoA substrate, indicative of a cooperative binding model. P053 reduces maximal reaction rate ($V_{max}$) without notably affecting substrate affinity ($K_{half}$) (Fig. 1c, d), indicating that it is a non-competitive inhibitor.

In a live-cell ceramide synthase assay[34] using cortical neuron cultures, a two hour pre-treatment with P053 resulted in dose-dependent inhibition of de novo C18:0 ceramide synthesis from deuterated dihydrosphingosine (Fig. 2a), confirming that the drug is cell permeable and inhibits CerS1 activity in living cells. We next tested the potency and specificity of our inhibitor in reducing endogenous ceramides. Most cell lines produce very little C18 ceramide. HEK293 cells produce more C18 ceramide than other immortalised cell lines that we tested, and we therefore quantified endogenous ceramides in HEK293 cells treated for 24 h with P053. C18:0 ceramide was reduced by 53% with 100 nM P053 (Fig. 2b, Supplementary Figure 1 and Supplementary Data File 1). No other forms of ceramide were reduced by P053 treatment, but a minor increase in C24:0 ceramide was observed (Fig. 2b). In addition to C18 ceramide, the compound selectively reduced C18 HexCer (Fig. 2c) and SM (Fig. 2d), which are both derived from C18 ceramide. P053 had no effect on HEK293 cell viability at or below 1 µM (Fig. 2e).

FTY720 has been reported to inhibit sphingosine kinase 1 in the low micromolar range[35]. P053 did not inhibit recombinant human sphingosine kinase 1 or 2 to any extent at 10 µM, whereas 10 µM FTY720 produced a 60% reduction in sphingosine kinase 1 activity (Supplementary Figure 2).

**P053 selectively targets CerS1 in vivo.** To investigate the in vivo efficacy of P053, bioavailability was tested by administering a single 5 mg/kg dose by oral gavage to mice (Fig. 3a). A maximal plasma concentration of 20 nM was achieved 4 h after administration, with a plasma half-life of 28 h. Expression profiling confirmed previous literature[18] and showed that CerS1 is highly expressed in brain and SkM, with minimal or no expression in other tissues, including adipose tissue (Fig. 3b). In a pilot study, 7 days administration of P053 at 5 mg/kg/day reduced C18 ceramide levels in SkM by 31%, whereas 1 mg/kg/day had no effect. To fully characterise the effects of the compound on the lipidome, we administered 5 mg/kg P053 daily for 4–6 weeks to mice fed either normal chow or a HFD. Of 302 individual lipid species manually verified following lipidomic profiling of SkM, only 18:1/18:0 and 18:2/18:0 ceramide were significantly reduced at $P$ < 0.01, after adjusting for multiple comparisons (Fig. 3c–e, and

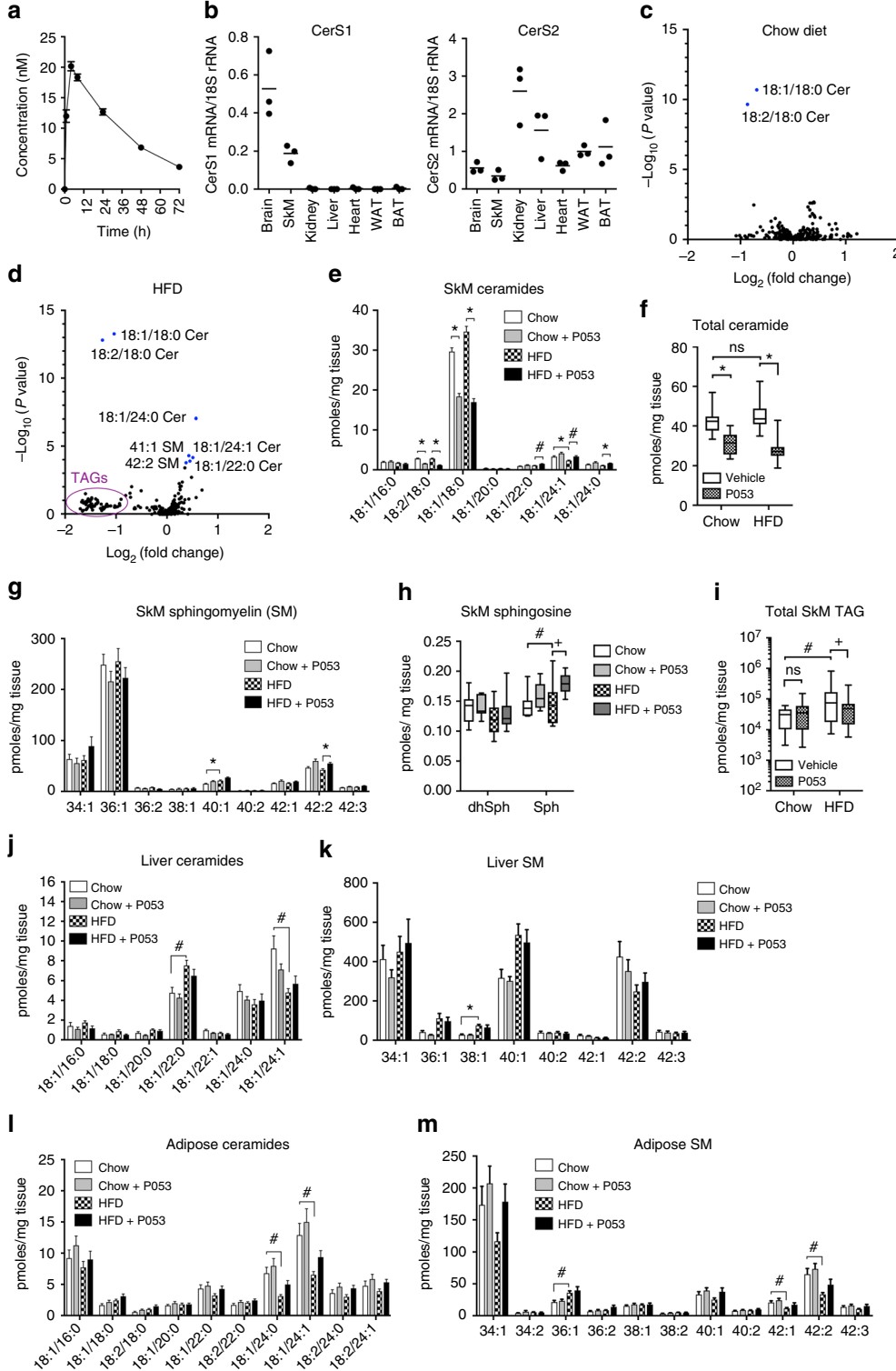

Supplementary Data File 2). As C18 is the dominant ceramide in SkM, P053 treatment reduced total ceramide content in this organ (Fig. 3f; $P = 0.73$ for effect of diet, $P < 0.001$ for effect of P053 by 2-way ANOVA). Very long chain ceramides (C22:0, C24:0 and C24:1) were significantly increased by P053 treatment in the HFD group (Fig. 3d, e), as was 42:2 SM, which is assumed to be the 18:1/24:1 (C24:1) form (Fig. 3g). Mean levels of 36:1 SM, which is derived from C18:0 ceramide, were 13% lower in P053-treated compared to vehicle-treated mice on both diets (Fig. 3g), however this effect was not statistically significant. HexCer

content was not significantly affected by P053 treatment (Supplementary Data File 2). The increase in C24 ceramides was not associated with increased C24 ceramide synthase activity (Supplementary Figure 3), and may therefore be a consequence of increased sphingoid base substrate availability due to the reduction in CerS1 activity. In accordance with this model, sphingosine levels were increased 22% as a result of CerS1 inhibition with P053 in mice fed a HFD (Fig. 3h; $P = 0.01$), and 11% in mice fed a chow diet (not significant). Dihydrosphingosine levels were not notably affected by P053 treatment (Fig. 3h).

**Fig. 3** CerS1 inhibition selectively reduces C18 ceramide in SkM. **a** Plasma P053 concentration following a single 5 mg/kg oral dose ($n = 5$ mice; mean ± SEM). **b** CerS1 mRNA expression was determined in a panel of mouse tissues by quantitative PCR ($n = 3$ mice; horizontal bar shows mean). Levels are normalised to 18S rRNA. WAT epididymal white adipose tissue, BAT brown adipose tissue. Results for CerS2 are also shown, to demonstrate the presence of RNA in samples from tissues lacking CerS1 expression. **c, d** Volcano plots showing P value plotted against fold change for each of 302 lipids in SkM of P053-treated relative to vehicle-treated mice. **c** Chow diet, **d** HFD, $n = 18$–20 per group. Lipids significantly altered in P053 vs vehicle are shown in blue and labelled by name. **e** Levels of individual ceramides, **f** total ceramide, **g** individual SM species, **h** dihydrosphingosine (dhSph) and sphingosine (Sph), and **i** total TAG, in SkM of mice fed chow or HFD with vehicle or 5 mg/kg P053 for 4–6 weeks ($n = 18$–20 per group; $n = 8$–10 per group for dhSph and Sph). **j**–**m** Liver ceramides (**j**), liver SM species (**k**), epididymal adipose ceramides (**l**), and epididymal adipose SM species (**m**); $n = 10$ per group. Bar graphs show mean ± SEM; #$P < 0.01$; *$P < 0.001$, as determined by two-tailed $t$-tests adjusted for multiple comparisons ($P$ values are after adjusting for multiple comparisons). Box and whisker plots show full data range with 25th—75th percentile boxed, and horizontal bar marking the median. Results in (**f**), (**h**), and (**i**) were analysed by two-way ANOVA, with Fisher's exact post-test to compare individual groups (+, $P < 0.05$; #$P < 0.01$; *$P < 0.001$). Raw data for **e**, **g**, **j**, **k**, **l** and **m** is provided in Supplementary Data File 1

**Table 2 Correlations between SkM ceramides and body fat**

| Ceramide | Body fat (%) | | WAT weight | |
|---|---|---|---|---|
| | r | P | r | P |
| Total | 0.010 | 0.54 | 0.138 | 0.40 |
| 18:1/16:0 | −0.156 | 0.34 | −0.164 | 0.31 |
| **18:1/18:0** | **0.367** | **0.020** | **0.393** | **0.012** |
| 18:1/20:0 | −0.197 | 0.22 | −0.030 | 0.85 |
| 18:1/22:0 | 0.209 | 0.196 | 0.267 | 0.096 |
| **18:1/24:0** | **−0.524** | **0.0005** | **−0.464** | **0.0025** |
| **18:1/24:1** | **−0.5353** | **0.0004** | **−0.428** | **0.0058** |

Spearman correlation co-efficients ($r$) and $P$ values are shown. Significant associations are in bold. Body fat % was determined by EchoMRI, whilst WAT weight is the combined weight of inguinal and epididymal adipose pads. Mice from both chow and HFD groups were used for this analysis ($n = 38$ mice). Mice treated with P053 were not included in this analysis, due to the confounding effect of P053 on muscle ceramides

This untargeted lipidomic data using quadriceps was supported by targeted analysis of ceramide, SM and HexCer in gastrocnemius muscle, which showed a significant increase in C18 ceramide content in the HFD compared to the chow control group, and significant reductions in C18 ceramide (i.e., 18:1/18:0), C18 dihydroceramide (i.e., 18:0/18:0), and 18:2/18:0 ceramide with P053 treatment (Supplementary Figure 4).

P053 treatment did not affect levels of common phospholipids from the phosphatidylcholine, phosphatidylethanolamine, phosphatidylserine, and phosphatidylinositol families (Supplementary Data File 2). We noted that the levels of most triacylglycerol (TAG) species were reduced by more than 50% in SkM of the HFD + P053 compared to HFD + vehicle group (Fig. 3d). Total SkM TAG content was increased five-fold in mice on a HFD compared to a chow diet (Fig. 3i; $P = 0.01$ for main effect of diet, $P = 0.15$ for effect of P053 by two-way ANOVA), and was 60% lower in the HFD + P053 group compared to the HFD vehicle group ($P = 0.021$). P053 treatment did not affect SkM TAG levels in mice fed normal chow (Fig. 3i). TAGs are synthesized directly from diacylglycerol (DG). DG levels were unaffected by P053 treatment (Supplementary Figure 5), suggesting that P053 does not directly inhibit TAG synthesis from DG.

Of 265 lipids measured in liver, none was significantly affected by P053 treatment in mice fed either chow or HFD (Supplementary Data File 2), despite the fact that there is substantial uptake of P053 in the liver (Supplementary Figure 6). There were 40 lipids significantly affected by diet ($P < 0.01$ after adjusting for multiple comparisons), including C22:0 and C24:1 ceramide (Fig. 3j), in line with previous results[20], and 38:1 SM (Fig. 3k). Similarly, P053 did not significantly impact on levels of C18 ceramide or SM, nor other ceramide or SM species, in adipose tissue, although there were a number of diet-induced changes (Fig. 3l, m). The lack of effect of P053 in liver and adipose tissue is likely a reflection of the very low levels of CerS1 in these tissues

(Fig. 3b). Although CerS1 is highly expressed in brain (Fig. 3b), P053 did not affect C18 ceramide or SM (particularly SM 36:1) levels in cerebellum (Supplementary Figure 7). This may be a consequence of three-fold lower levels of the compound in brain tissue compared to SkM (Supplementary Figure 6).

The action of FTY720 as an immunosuppressant is dependent on its phosphorylation[36,37]. AAL(S), from which P053 is derived, is a non-phosphorylatable FTY720 analogue[37]. In cultured HEK293 cells incubated with FTY720, FTY720-phosphate was clearly detected, whereas the phosphate of P053 was not detected in either cultured cells or plasma of mice administered the drug, indicating that like the parent compound AAL(S), P053 is non-phosphorylatable (Supplementary Figure 8).

**CerS1 inhibition prevents fat deposition but not insulin resistance induced by a HFD.** A recent study showed that the C18 ceramide content of SkM is positively correlated with visceral fat mass in humans[22]. In vehicle-treated mice in the current study, C18 ceramide levels in SkM were positively correlated, and C24 ceramides were inversely correlated, with body fat (Table 2). These correlations are indicative of a relationship between the ceramide content of SkM and whole-body adiposity. In agreement with this hypothesis, P053 treatment significantly reduced whole-body fat mass and the weight of individual white adipose depots in mice fed a HFD (Fig. 4a–e). Mean fat pad weights were 0.52 g (epididymal) and 0.27 g (inguinal) higher in mice fed a HFD compared to a chow diet. This gain in fat mass was almost halved to 0.29 g (epididymal) and 0.15 g (inguinal) with P053 treatment. This reduced fat mass was not due to any effect of P053 on food intake (Supplementary Figure 9). There is a close association between excess adiposity and insulin resistance[38] and as expected HFD-fed mice displayed elevated fasting insulin (Fig. 4f) and became glucose intolerant (Fig. 4g) due to impaired glucose disposal during the glucose tolerance test (Supplementary Figure 10). Hyperinsulinemic/euglycemic clamps revealed that fat-fed animals displayed a lower glucose infusion rate (indicating whole-body insulin resistance) (Fig. 4h), impaired peripheral glucose disposal (Fig. 4i) and reduced insulin-stimulated glucose uptake in SkM (quadriceps and soleus; Fig. 4j, k). Despite its marked effects on adiposity and SkM ceramide, P053 treatment had no impact on any of these measures of glucose homoeostasis and insulin action (Fig. 4f–k).

**CerS1 inhibition primes SkM to oxidise fatty acids.** In simple terms, lipid accumulation in tissues is determined by the balance between oxidation and storage. Previous studies have reported that the lipotoxic effects of C16 ceramide are partly due to inhibition of fatty acid β-oxidation in liver and adipose tissue[9,10]. Muscle homogenates from chow and HFD-fed mice treated with P053 displayed increased oxidation of $^{14}$C-palmitate (Fig. 5a), suggesting that enhanced lipid oxidation may underlie the reduced lipid accretion in response to P053. To more directly

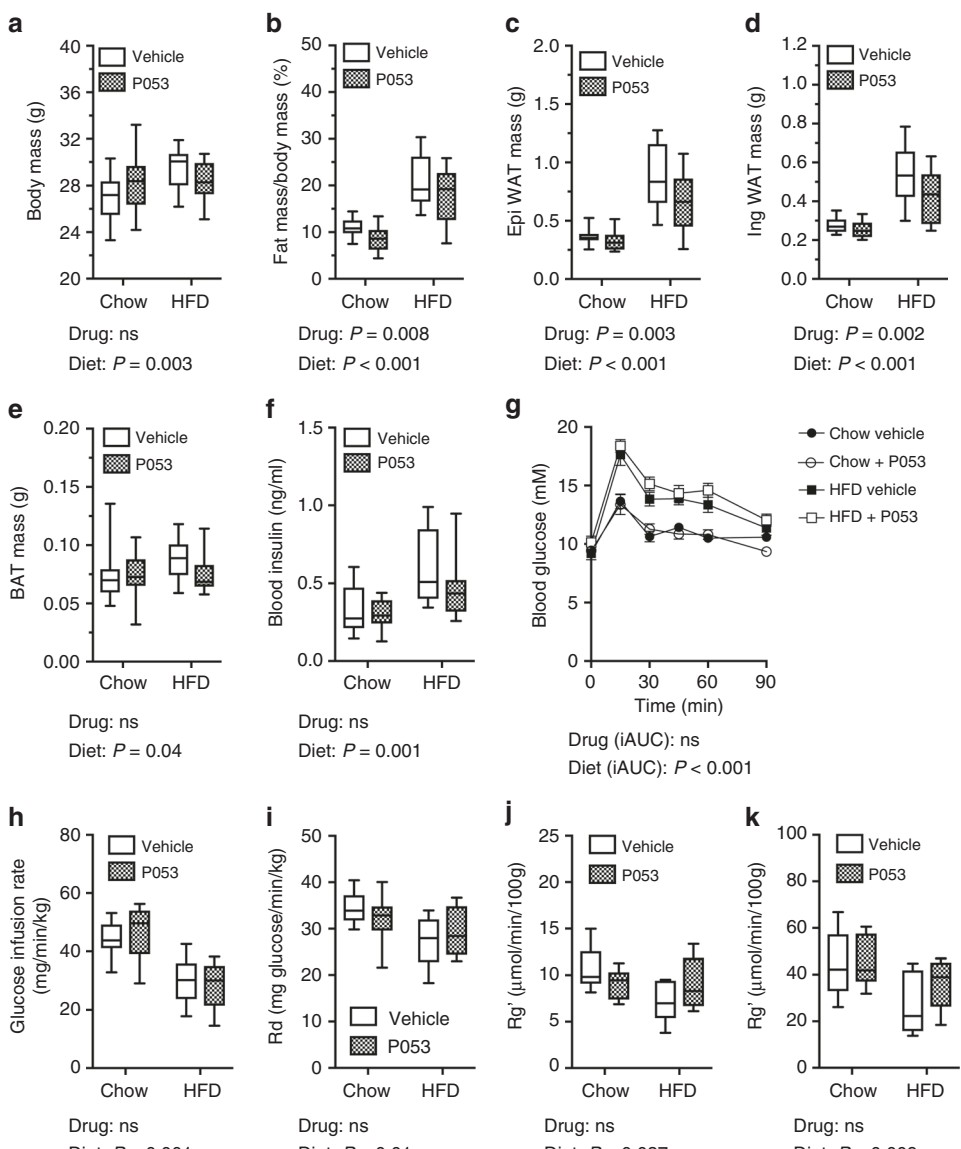

**Fig. 4** CerS1 inhibition impedes fat accumulation but not insulin resistance in mice fed a HFD. **a** Total body mass, **b** fat mass as a % of body mass determined by EchoMRI, **c** epididymal white adipose tissue (WAT) mass, **d** inguinal WAT mass, and **e** brown adipose tissue (BAT) mass, in mice fed chow or HFD with vehicle or 5 mg/kg P053 for 4–6 weeks ($n = 20$ per group). **f** Fasting blood insulin and **g** blood glucose following an oral glucose load in mice fed chow or HFD with vehicle or 5 mg/kg P053 ($n = 10$ per group; mean ± SEM). **h** Glucose infusion rate, **i** rate of glucose disposal from the circulation (Rd), and **j**, **k** rate of glucose uptake (Rg') into quadriceps (**j**) and soleus (**k**) muscles, as determined in hyperinsulinemic-euglycemic clamps ($n = 6$–9 mice per group as detailed in the methods). Box and whisker plots (**a**–**f**) show full data range with 25th—75th percentile boxed, and horizontal bar marking the median. P-values showing the main effect of diet or drug, as determined by two-way ANOVA, are shown below each graph; ns not significant. ANOVA results for the incremental Area Under the Curve (iAUC) are given for (**g**)

differentiate between inhibition of TAG synthesis and enhanced lipid oxidation, we quantified TAG synthesis from $^{14}$C-palmitate, as well as production of $CO_2$ and acid soluble metabolites (ASM) (i.e., β-oxidation intermediates), in soleus muscles from mice treated for 2 weeks with P053. Incorporation of $^{14}$C-palmitate into TAG during the 1 h incubation was unchanged (Fig. 5b), but oxidation of $^{14}$C-palmitate was significantly increased in soleus muscles from P053-treated mice (Fig. 5c). To assess if this was an acute effect of the compound, isolated soleus muscles were treated in vitro for 1.5 h with P053 before the assay. Palmitate oxidation was not affected by acute P053 treatment (Fig. 5d), suggesting that prolonged CerS1 inhibition is required to prime SkM to partition fatty acids to oxidative pathways over storage.

Fatty acid oxidation occurs in mitochondria and ceramides have been reported to directly influence mitochondrial function, as well

as mitochondrial morphology and turnover[9,19,39]. Assessment of mitochondrial respiration in permeabilised muscle fibres revealed that the enhanced lipid oxidation in SkM in response to CerS1 inhibition was likely due to an overall increase in mitochondrial capacity, as there was a significant increase in respiratory activity in the presence of complex I and IV substrates, and a trend for an increase in complex II activity (Fig. 5e). Increased respiratory capacity was associated with significant up-regulation of genes encoding both mitochondrial-encoded (Co-2, Cytb, Atp6) and nuclear-encoded (Ndufb5, Atp5o, Cycs, Cox5b) respiratory complex subunits (Fig. 6a). Respiratory complexes were also up-regulated at the protein level in SkM of P053-treated mice (Fig. 6b, c), accompanied by increased activity of the TCA cycle enzyme citrate synthase—a commonly used marker of mitochondrial content—and increased activity of the β-oxidation enzyme βHAD

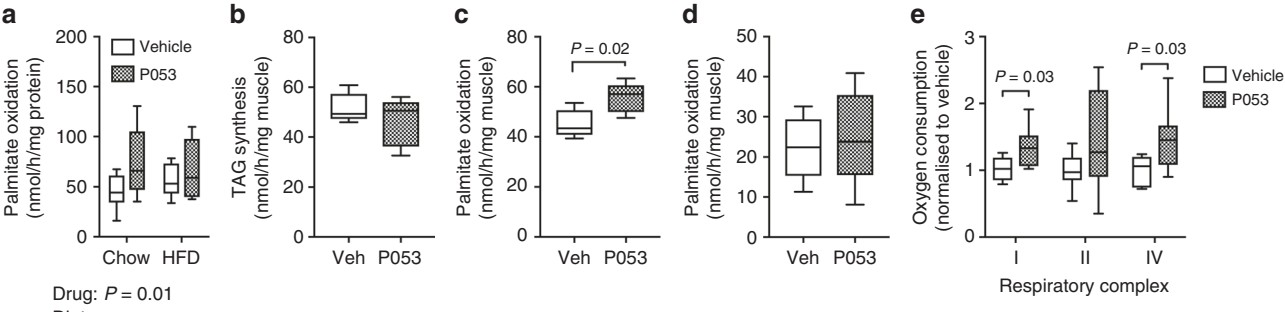

**Fig. 5** CerS1 inhibition increases palmitate oxidation and mitochondrial respiratory capacity in SkM. **a** $^{14}$C-palmitate oxidation in SkM homogenates from mice treated for 6 weeks with P053 or vehicle ($n = 10$ mice). **b** $^{14}$C-palmitate incorporation into TAG, and **c** $^{14}$C-palmitate oxidation, in isolated soleus muscles taken from mice treated for two weeks with 5 mg/kg P053 or vehicle control ($n = 5$ mice). Results show nmoles $^{14}$C-palmitate converted to **b** TAG or **c** $CO_2$+acid soluble metabolites. **d** Palmitate oxidation in soleus muscle strips pre-treated for 1.5 h in vitro with 1 µM P053 or vehicle control ($n = 5$ mice). **e** ADP-stimulated (State 3) respiratory complex activities measured in permeabilised extensor digitorum longus muscle fibres of mice treated for three weeks with 5 mg/kg P053 or vehicle control ($n = 7$ mice). Statistical significance in (**a**) was determined by two-way ANOVA (main effect of drug and diet reported below the graph), and in (**b–e**) by two-tailed t-tests (results where $P < 0.05$ are indicated on the graph)

($P = 0.06$) (Fig. 6d). This effect was specific to muscle, with no increase in mitochondrial markers in liver (Fig. 7a–c), where CerS1 and C18:0 ceramide are low, and P053 had no significant effect on lipid levels.

Increased transcription of respiratory complex subunits suggested that P053 enhances mitochondrial biogenesis. Consistent with this, P053 treatment significantly increased protein expression of Pgc-1α, the master regulator of mitochondrial biogenesis[40] (Fig. 6e) and elevated mitochondrial DNA content (Fig. 6f; $P = 0.06$ for main effect of P053 in 2-way ANOVA; $P = 0.007$ for post-test comparing HFD vehicle to HFD + P053). The expression of transcriptional regulators of mitochondrial biogenesis Gabp1α (Nuclear respiratory factor 2)[41] and Nfe2l2 (nuclear factor (erythroid-derived 2)-like 2)[42] was also significantly increased by P053 treatment (Fig. 6g). AMP-activated protein kinase (AMPK) is an important regulator of mitochondrial biogenesis in muscle[43] whose activity is influenced by intracellular ceramide levels[44]. P053 treatment had no effect on phosphorylation of AMPK or its downstream substrate acetyl-CoA carboxylase (ACC) (Supplementary Figure 11), suggesting that this pathway is not involved in the enhanced mitochondrial function induced by CerS1 inhibition.

## Discussion

There is a strong body of evidence indicating that different forms of ceramide, synthesized by different CerS isoforms, regulate distinct physiological processes[1,8]. However, research on the role of specific CerS isoforms is hampered by the absence of isoform-selective CerS inhibitors. Researchers seeking to pharmacologically inhibit ceramide synthesis in vivo have generally used myriocin[45–47], which inhibits serine palmitoyltransferase, the rate limiting initial step in the biosynthesis of all sphingolipids. This report describes the first potent and selective inhibitor of any CerS isoform, facilitating the discovery of a physiological role for CerS1 in the regulation of mitochondrial function and fatty acid metabolism. P053 exhibits strong selectivity for CerS1 over other CerS isoforms, potently and very specifically reducing C18 ceramide levels in cultured cells and mouse SkM. CerS1 inhibition with P053 was also successfully dissected from non-specific pro-apoptotic properties of sphingoid base analogues[27,48,49], as the compound inhibited C18 ceramide synthesis at concentrations at least one order of magnitude lower than those required to promote apoptosis in HEK293 cells. Interestingly, the mode of inhibition was non-competitive with respect to either the

sphingosine or C18 fatty acyl-CoA substrate, suggesting that P053 may bind an allosteric site, possibly a lipid binding site on the enzyme. In the absence of structural data for the ceramide synthase family, or sitable homology models, we are unable to fully characterise the mechanism of inhibition. Both CerS1 and CerS4 activities were assayed using C18:0 fatty acid substrate, indicating that inhibition by P053 is specific to the CerS1 isoform and not the specific fatty acid substrate used to measure enzyme activity.

Our report provides the first pharmacological evidence that endogenous CerS1 is indeed highly selective for C18 fatty acid substrates, as treatment of cells or mice with P053 resulted in selective reduction of C18 ceramide. These findings support experiments in which CerS1 has been overexpressed[15], silenced[50], or knocked out[14]. We observed small but statistically significant increases in C24 ceramides in SkM of mice following inhibition of CerS1. This was not associated with increased C24 ceramide synthase activity and is therefore most likely a consequence of increased availability of sphingoid base substrates for CerS2-catalysed C24 ceramide synthesis in the absence of CerS1 activity. This is consistent with two distinct genetic mouse models of CerS1 deficiency[14,51], and supported by our observation that sphingosine levels were only modestly increased, and dihydrosphingosine levels were not increased, in SkM of P053-treated mice. C22 and C24 ceramide levels are an order of magnitude higher than free sphingosine in SkM, so re-routing of sphingoid base substrates into these ceramides would prevent significant increases in sphingosine upon CerS1 inhibition. In both HEK293 cells and mouse SkM the reduction in C18 forms of SM were much less prominent than the changes in C18 ceramides. These findings indicate that either the turnover rate for SM is slower than ceramide, or that steady state C18 SM levels are preferentially maintained at the expense of C18 ceramide.

P053 treatment significantly affected ceramide content, particularly C18 ceramide, in SkM, but not liver or adipose tissue. This tissue-specific pattern may simply be a reflection of the substantial differences in CerS1 expression/activity between SkM and other organs. Although CerS1 is highly specific for C18 ceramide synthesis, other CerS isoforms, particularly CerS4 and 5, are capable of catalysing C18 ceramide synthesis[12,16]. However, it is also possible that lower abundance ceramide species may be primarily obtained from the circulation and not through endogenous synthesis in some organs. In support of this notion, short-term infusion of LDL containing C24:0 ceramides in mice was shown to alter membrane C24:0 ceramide content in SkM[52]. Surprisingly, there was no effect of P053 on ceramide levels in the

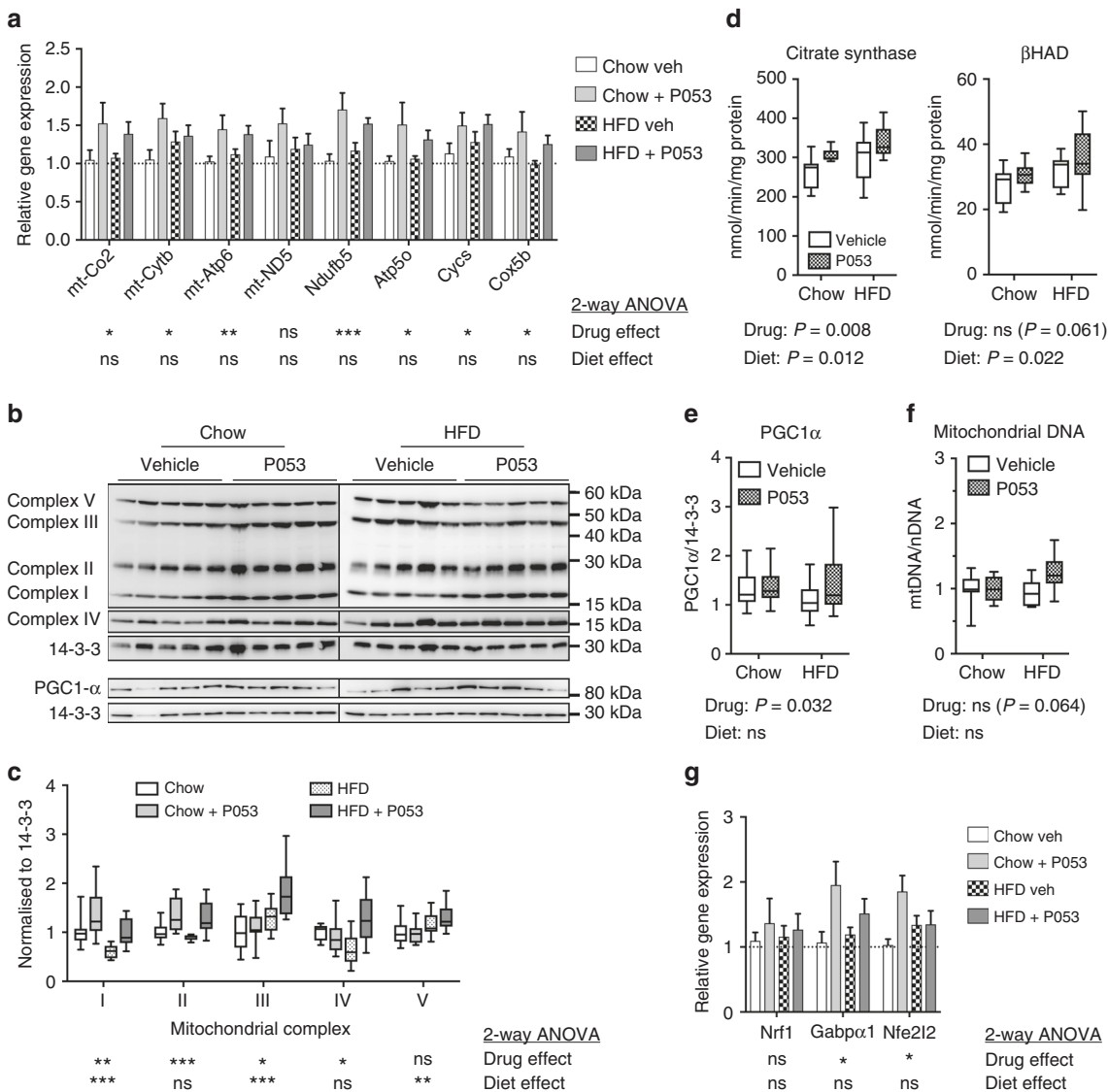

**Fig. 6** CerS1 inhibition increases mitochondrial markers in SkM. **a** Levels of mitochondrially (mt)- and nuclear-encoded respiratory complex subunits in SkM, as determined by real-time PCR; $n = 9$ for chow diet groups; $n = 10$ for HFD groups. **b** Western blots of respiratory complex subunits in SkM. **c** Densitometry for respiratory complex subunits relative to 14-3-3 protein. **d** Citrate synthase and β-hydroxyacyl coenzyme A dehydrogenase (βHAD) activity in SkM. **b**–**d**, $n = 10$ mice for all groups except the chow + P053 group, for which $n = 8$. **e** Western blot of PGC-1α protein levels with densitometry normalised to 14-3-3 protein, $n = 18$–$20$ mice per group. **f** Mitochondrial DNA (mtDNA) content, normalised to nuclear DNA (nDNA) and expressed relative to the mean of the control group; $n = 10$ mice for all groups except the chow + P053 group, for which $n = 8$. **g** Gene expression levels for regulators of mitochondrial biogenesis and function, as determined by real-time PCR; $n = 8$ for chow diet groups; $n = 10$ for HFD groups. **a**, **g** show mean ± SEM. Box and whisker plots show full data range with 25th–75th percentile boxed, and horizontal bar marking the median. Statistical significance was determined by two-way ANOVA, with the main effect of the drug or diet shown beneath each result; *$P < 0.05$; **$P < 0.01$; ***$P < 0.001$. Raw data for panels **a** and **g** is provided in Supplementary Data File 1

brain, despite high CerS1 expression levels. This appeared to be due to low uptake and/or retention of P053 in this organ (Supplementary Figure 6), but could also indicate a lower rate of sphingolipid turnover in brain compared to SkM. Genetic ablation of CerS1 causes neurodegeneration and cerebellar atrophy during development[14,51], and the lack of an effect of P053 on brain ceramides indicates that treatment with P053 is unlikely to cause neuronal atrophy.

Our CerS1 inhibitor empowered us to dissect the role for SkM C18 ceramide in fat metabolism from its proposed role in insulin sensitivity. C18 ceramide levels were 20–30% higher in SkM of mice fed a high fat compared to chow diet. CerS1 inhibition with P053 significantly impeded fat deposition in mice fed a HFD and

reduced the C18 ceramide level in SkM to well below that of control mice, but did not have any impact on glucose disposal, nor insulin-stimulated glucose uptake into SkM. Blocking total sphingolipid synthesis with myriocin has been shown to protect against insulin resistance and hepatic steatosis in rodents[45–47], as has genetic ablation of Cers5 and CerS6 in liver and adipose tissue[10]. It is therefore possible that CerS isoforms that are more highly expressed in liver and adipose tissue, specifically CerS2, CerS5, and CerS6, play a much more significant role than CerS1 in regulating glucose homoeostasis and insulin action. This, in-turn, suggests either that the associations between C18 ceramide and insulin resistance in humans[11,22,23] are correlative, but not causal; or that C18 ceramide in muscle is a more significant

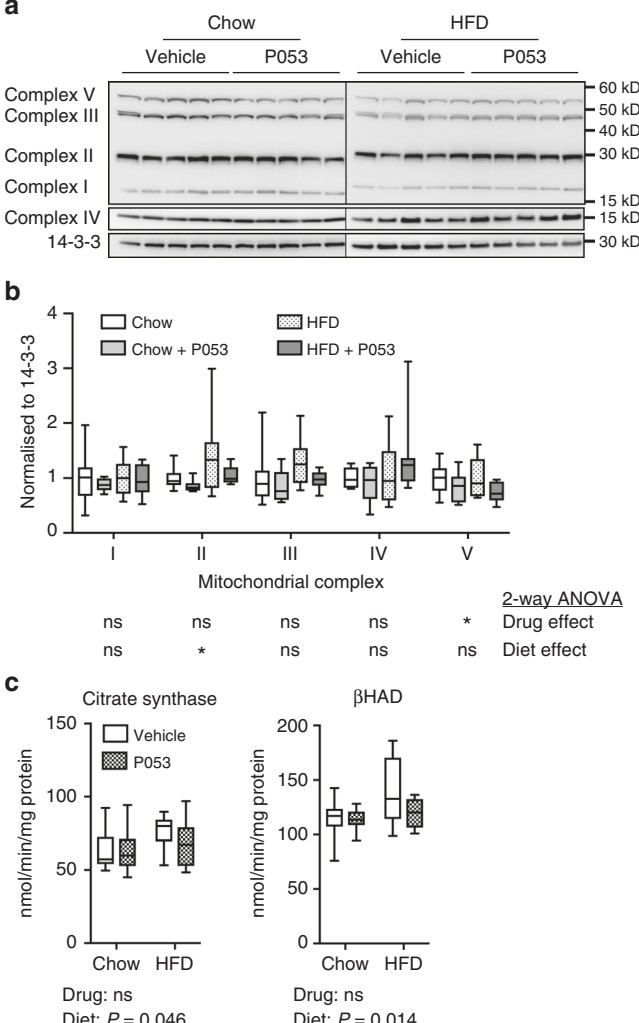

**Fig. 7** CerS1 inhibition does not affect mitochondrial markers in liver. **a** Western blots of respiratory complex subunits, **b** densitometry for respiratory complex subunits relative to 14-3-3 protein, and **c** Citrate synthase and βHAD activity in liver homogenates; $n = 10$ mice for all groups. Box and whisker plots show full data range with 25th–75th percentile boxed, and horizontal bar marking the median. Statistical significance was determined by two-way ANOVA, with the main effect of the drug or diet shown beneath each result; *$P < 0.05$

provides a rational explanation for the reduced fat deposition in P053-treated mice on a HFD. However, to the best of our knowledge, there are no prior studies demonstrating that specifically enhancing fat metabolism in SkM reduces adiposity in mice fed a HFD. In fact, previous studies have shown that genetically enhancing fatty acid oxidation in SkM, through deletion of acetyl-CoA carboxylase 2 or muscle-specific expression of carnitine palmitoyltransferase-1, did not reduce adiposity[54–56]. An important point of difference with our study is that increased fatty acid oxidation was coupled with elevated markers of mitochondrial content and activity, i.e., CerS1 inhibition appears to have increased overall mitochondrial capacity and not just altered fuel selection. The rate of palmitate oxidation in isolated SkM was directly comparable to its incorporation into TAG (Fig. 5). Over days to weeks the sustained increase in fatty acid oxidation with P053 may be sufficient to offset fat deposition and weight gain. Additional evidence using other approaches to boost mitochondrial content and fatty acid oxidation is required to further test this possibility.

The precise mechanism through which chronic CerS1 inhibition changes mitochondrial content and/or function requires further investigation. Our findings of elevated expression of PGC-1α and other important transcriptional regulators of mitochondrial content and function (Gabp1α[41] and Nfe2l2[42]) suggest enhanced mitochondrial biogenesis in response to depletion of C18 ceramide in muscle. Other studies have shown down-regulation of genes regulating fatty acid metabolism in the absence of CerS2[57], associated with reduced β-oxidation of fatty acids[9]. These effects were attributed to elevation of C16 ceramide in the absence of very long chain ceramide synthesis. Similarly, loss of CerS2 was shown to reduce respiratory enzyme activities and increase mitochondrial oxidative stress, effects that were also attributed to elevated C16 ceramide[58]. Our data indicates that C18 ceramide behaves similarly to C16 ceramide in suppressing respiratory enzyme levels and fatty acid β-oxidation. Another possible contributor to changes in mitochondrial content/function in response to P053 treatment is reduced mitophagy, since exogenous CerS1 expression and a mitochondrial-targeted analogue of C18 ceramide were both shown to promote mitophagy[39]. Ceramides have also been shown to induce mitochondrial fission in SkM cells, reducing their oxidative capacity[19]. Thus, another possibility is that CerS1 inhibition, by lowering the overall ceramide content of SkM, boosts respiratory capacity via inhibition of fission.

There are several lines of evidence in the literature indicating that ceramides regulate fat storage. Firstly, genetic ablation of the *Drosophila* CerS homologue *Schlank* results in the absence of fat pads[59]. Secondly, the serine palmitoyltransferase inhibitor myriocin reduces adiposity and hepatic steatosis in rodents fed a HFD[45–47]. Our finding that CerS1 inhibition significantly impedes fat deposition is perhaps surprising given that CerS1 is not expressed to a significant extent in adipose tissue or liver. However, C18 ceramide in SkM was positively correlated with visceral fat mass, blood pressure, and liver fat in a recent study in humans[23], and we identified a significant positive correlation between C18 ceramide in SkM and whole body fat mass in mice, as determined by two different methods: Echo MRI and weight of dissected fat pads (Table 2). Our experiments with P053 suggest that this is a causal relationship, as P053 significantly reduced C18 ceramide in SkM without affecting ceramide levels in liver and adipose tissue. We note that an even stronger inverse correlation of C24 ceramides with adipose tissue weight and % body fat was observed. The inhibition of CerS1 may therefore reduce adiposity both directly by reducing C18 ceramide in SkM and indirectly through the associated increase in C24 ceramide. We cannot definitively rule out a direct effect of CerS1 inhibition on

contributor to insulin resistance in humans exposed to a mixed diet of fats and simple carbohydrates than to rodents on a defined HFD.

Our results highlight a new role for CerS1 as a key regulator of mitochondrial oxidative capacity, with chronic CerS1 inhibition priming SkM to metabolise fatty acids. Palmitate oxidation was increased in both isolated muscles and SkM homogenates of mice treated with P053, but not following acute exposure to the drug. The enhanced palmitate oxidation following CerS1 inhibition was associated with higher respiratory capacity, increased mitochondrial protein levels, and increased gene expression for respiratory complex subunits. The dependence of these effects on CerS1 was evident in the fact that P053 treatment affected mitochondrial markers in SkM but not liver, where CerS1 levels are very low. Given the substantial contribution of SkM to whole-body energy expenditure[53], enhanced channeling of fatty acids into oxidative pathways over storage in muscle over the course of the study

adipose tissue causing reduced adiposity, however, this seems unlikely given the absence of CerS1 in this tissue and the fact that C18 and C24 ceramide levels in adipose tissue were not affected by our inhibitor.

In summary, we have generated the first isoform-selective CerS inhibitor, specifically targeting CerS1 with nanomolar potency. Our results support a model in which CerS1 activity and C18 ceramide in SkM are not requisite for the development of insulin resistance in mice fed a HFD, but do play a very significant role in storage of dietary fats by acting as a brake on mitochondrial fatty acid oxidation. Our inhibitor will provide a valuable resource for researchers seeking to understand the functions of CerS1 in physiology and pathology, and the marked influence of P053 on whole-body and tissue-specific lipid accumulation further strengthens the notion that targeting ceramide synthesis may be a viable therapeutic option for treating obesity.

## Methods

**Synthesis and chemical characterisation of GO24 and P053 (2R,5S)-5-Iso-propyl-3,6-dimethoxy-2-(4′-benzyloxyphenethyl)-2,5-dihydropyrazine.** A solution of *n*-butyllithium in hexanes (1.7 M, 6.0 mL, 10.25 mmol) was added dropwise to a solution of freshly distilled (S)-Schöllkopf's reagent (1.89 g, 10.26 mmol) in freshly distilled THF (22 mL) at −78 °C (dry ice/acetone). The solution was stirred at −78 °C for 15 min, where it had turned dark yellow. A solution of 2-(4′-benzyloxyphenyl)-1-iodoethane[60] (3.50 g, 10.35 mmol) in freshly distilled THF (21 mL) at −78 °C was added dropwise. The solution was stirred for a further 30 min at −78 °C then allowed to slowly warm to −15 °C over 4 h. The reaction mixture was quenched with saturated aqueous sodium bicarbonate solution and allowed to warm to room temperature. The THF was removed under reduced pressure and the residue extracted with dichloromethane (×4). The organic extracts were combined and dried ($Na_2SO_4$). The solvent was removed under reduced pressure and the crude material was purified by flash chromatography on silica gel, eluting with 3% ethyl acetate/*n*-hexane, to afford the product as a clear colourless oil (3.84 g, *quant*). $[\alpha]_D^{25.0°C} = -6$ (0.5, $CHCl_3$); [1]H NMR (300 MHz; $CDCl_3$) δ 0.70 (d, *J* = 6.8 Hz, 3 H), 1.06 (d, *J* = 6.8 Hz, 3H), 1.92–2.02 (m, 1H), 2.07–2.19 (m, 1H), 2.22–2.32 (m, 1H), 2.48–2.64 (m, 1H), 3.69 (s, 3H), 3.70 (s, 3H), 3.97 (t, *J* = 3.4 Hz, 1H), 4.02–4.07 (m, 1H), 5.04 (s, 2H), 6.89 (dd, *J* = 2.1, 4.7 Hz, 2H), 7.10–7.12 (m, 2H), 7.29–7.45 (m, 5H); [13]C NMR (75 MHz; $CDCl_3$) δ 16.8, 19.2, 30.2, 32.0, 36.2, 52.5, 55.1, 60.9, 70.2, 114.8, 127.6, 128.0, 128.7, 129.5, 134.7, 137.4, 157.1, 163.7, 163.9; IR (NaCl, neat) 1694 cm$^{-1}$; HRMS (ESI-MS): *m/z* calcd for $C_{24}H_{31}N_2O_3$ [M + H]$^+$ 395.2334, found 395.2320.

**(2S,5S)-5-isopropyl-3,6-dimethoxy-2-(4′-benzyloxyphenethyl)-2-methyl-2,5-dihydropyrazine.** A solution of *n*-butyllithium in hexanes (1.7 M, 5.8 mL, 9.86 mmol) was added dropwise to a solution of (2R,5S)-5-isopropyl-3,6-dimethoxy-2-(4′-benzyloxyphenethyl)-2,5-dihydropyrazine (3.33 g, 9.38 mmol) in freshly distilled THF (47 mL) at −78 °C (dry ice/acetone). The solution was stirred at −78 °C for 15 min, where it had turned dark yellow. Methyl iodide (1.17 mL, 18.79 mmol) was added dropwise. The solution was stirred for a further 30 min at −78 °C then allowed to slowly warm to −15 °C over 4 h. The reaction mixture was quenched with saturated aqueous sodium bicarbonate solution and allowed to warm to room temperature. The THF was removed under reduced pressure and the residue extracted with dichloromethane (×4). The organic extracts were combined and dried ($Na_2SO_4$). The solvent was removed under reduced pressure and the crude material was purified by flash chromatography on silica gel, eluting with 2% ethyl acetate/*n*-hexane, to afford the product as a clear colourless oil (1.55 g, 45 %), with all analytical data matching that reported in the literature[27]. $[\alpha]_D^{25.0°C} = -6$ (0.5, $CHCl_3$); [1]H NMR (300 MHz; $CDCl_3$) δ 0.70 (d, *J* = 6.8 Hz, 3H), 1.12 (d, *J* = 6.8 Hz, 3H), 1.30 (s, 3H), 1.85 (td, *J* = 5.1, 7.7 Hz, 1H), 2.08 (td, *J* = 4.3, 8.6 Hz, 1 H), 2.25 (td, *J* = 4.3, 8.8 Hz, 1H), 2.34–2.48 (m, 2H), 3.70 (s, 3H), 3.71 (s, 3H), 3.93 (d, *J* = 3.3 Hz, 1H), 5.04 (s, 2H), 6.87–6.90 (m, 2H), 7.06–7.09 (m, 2H), 7.29–7.44 (m, 5H); [13]C NMR (75 MHz; $CDCl_3$) δ 17.1, 19.7, 28.7, 30.7, 30.8, 42.9, 52.4, 58.4, 60.5, 70.2, 114.8, 127.6, 128.0, 128.7, 129.4, 135.2, 137.4, 157.0, 162.1, 165.7; IR (NaCl, neat) 1694 cm$^{-1}$; HRMS (ESI-MS): *m/z* calcd for $C_{25}H_{33}N_2O_3$ [M + H]$^+$ 409.2491, found 409.2490.

**Methyl (2S)-2-amino-4-(4′-benzyloxyphenyl)-2-methylbutanoate.** A solution of trifluoroacetic acid (15 mL, 0.20 mol) in water (30 mL) was added dropwise to a solution of (2S,5S)-5-isopropyl-3,6-dimethoxy-2-(4′-benzyloxyphenethyl)-2-methyl-2,5-dihydropyrazine (1.48 g, 4.01 mmol) in acetonitrile (100 mL). The solution was stirred at room temperature for 4 h after which the acetonitrile was removed under reduced pressure. The residue was diluted with water and neutralised with portions of solid sodium bicarbonate, then extracted with dichloromethane (×4). The organic extracts were combined and dried ($Na_2SO_4$). The solvent was removed under reduced pressure and the crude material was purified

by flash chromatography on silica gel, eluting with ethyl acetate, to afford the product as a clear colourless oil (0.90 g, 72%), with all analytical data matching that reported in the literature[27]. $[\alpha]_D^{23.5°C} = +12$ (0.5, $CHCl_3$); [1]H NMR (300 MHz; $CDCl_3$) δ 1.37 (s, 3H), 1.80–1.90 (m, 1H), 1.96–2.06 (m, 1H), 2.41–2.51 (m, 1H), 2.55–2.65 (m, 1H), 3.70 (s, 3H), 5.04 (s, 2H), 6.88–6.91 (m, 2H), 7.07–7.10 (m, 2H), 7.31–7.44 (m, 5H); [13]C NMR (75 MHz; $CDCl_3$) δ 26.6, 29.9, 43.1, 52.3, 57.8, 70.1, 114.9, 127.5, 127.9, 128.6, 129.3, 134.0, 137.2, 157.1, 178.0; IR (NaCl, neat) 1733 cm$^{-1}$; HRMS (ESI-MS): *m/z* calcd for $C_{19}H_{24}NO_3$ [M + H]$^+$ 314.1756, found 314.1754.

**(2S)-2-amino-4-(4′-benzyloxyphenyl)-2-methyl-1-butanol (GO24).** Lithium aluminium hydride (0.15 g, 3.95 mmol) was added as a solid in one portion to a solution of methyl (2S)-2-amino-4-(4′-benzyloxyphenyl)-2-methylbutanoate (0.83 g, 2.64 mmol) in freshly distilled THF (40 mL) at 0 °C. The solution was stirred at 0 °C for 20 min then the cold bath was removed and the solution stirred at room temperature for 1 h. The reaction was quenched with saturated aqueous sodium sulphate solution and the mixture was extracted with ethyl acetate (×4). The organic extracts were combined and washed with saturated aqueous sodium bicarbonate solution, water and brine, and dried ($Na_2SO_4$). The solvent was removed under reduced pressure and the crude material was purified by flash chromatography on silica gel, eluting with 2% methanol/4% triethylamine/dichloromethane, to afford the product as a white solid (0.54 g, 72%), with all data matching that reported in the literature[27]. $[\alpha]_D^{25.5°C} = -2$ (0.5, $CHCl_3$); M.p.: 196.7–197.6 °C; [1]H NMR (300 MHz; $CDCl_3$) δ 1.13 (s, 3H), 1.61–1.71 (m, 2H), 2.59 (t, *J* = 8.6 Hz, 2H), 3.34 (dd, *J* = 7.5, 10.3 Hz, 1H), 5.04 (s, 2H), 6.90 (dd, *J* = 2.1, 4.6 Hz, 2H), 7.09–7.12 (m, 2H), 7.29–7.45 (m, 5H); [13]C NMR (75 MHz; $CDCl_3$) δ 24.6, 29.5, 42.2, 53.1, 70.2, 70.3, 115.0, 127.6, 128.0, 128.7, 129.3, 134.8, 137.3, 157.1; IR (NaCl, neat) 3425 cm$^{-1}$; HRMS (ESI-MS): *m/z* calcd for $C_{16}H_{28}NO_4$ [M + H]$^+$ 286.1807, found 286.1807.

**(2S)-t-butyl(1-hydroxy-4-(4′-benzyloxyphenyl)-2-methylbutan-2-yl)carbamate.** Di-*t*-butyl dicarbonate (0.42 g, 1.94 mmol) was added as a solid in one portion to a mixture of (2S)-2-amino-4-(4′-benzyloxyphenyl)-2-methyl-1-butanol (0.36 g, 1.28 mmol) in saturated aqueous sodium bicarbonate solution (12.8 mL) and ethyl acetate (12.8 mL). The mixture was heated at 70 °C for 16 h then allowed to cool to room temperature. The solution was diluted with water and extracted with ethyl acetate (×3). The organic extracts were combined and washed with brine, then dried ($Na_2SO_4$). The solvent was removed under reduce pressure and the crude material was purified by flash chromatography on silica gel, eluting with 30% ethyl acetate/*n*-hexane, to afford the product as a white solid (0.51 g, *quant*), with all analytical data matching that reported in the literature[61]. $[\alpha]_D^{25.6°C} = +2$ (0.5, $CHCl_3$); [1]H NMR (400 MHz; $CDCl_3$) δ 1.22 (s, 3H), 1.44 (s, 9H), 1.80–1.88 (m, 1H), 1.99–2.06 (m, 1H), 2.53 (td, *J* = 7.7, 12.2 Hz, 1H), 2.63 (td, *J* = 5.2, 12.0 Hz, 1H), 3.62–3.72 (m, 2H), 4.06 (br s, 1H), 4.62 (s, 1H), 5.04 (s, 2H), 6.89–6.91 (m, 2H), 7.10–7.12 (m, 2H), 7.30–7.44 (m, 5H).

**(2S)-t-butyl(1-hydroxy-4-(4′-hydroxyphenyl)-2-methylbutan-2-yl)carbamate.** Palladium on carbon (44 mg, 10 wt. %) was added to a solution of (2S)-*t*-butyl(1-hydroxy-4-(4′-benzyloxyphenyl)-2-methylbutan-2-yl)carbamate (0.44 g, 1.15 mmol) in dry methanol (30 mL) at room temperature. A hydrogen balloon was attached to the flask and the flask was evacuated and purged with hydrogen (×3). The solution was stirred at room temperature for 18 h. The reaction mixture was diluted with dichloromethane and filtered through a short pad of Celite, eluting with dichloromethane. The solvent was removed under reduced pressure to afford the product as a white solid (0.34 g, *quant*), with all the analytical data matching that reported in the literature[61]. $[\alpha]_D^{25.6°C} = +5$ (0.5, $CHCl_3$); [1]H NMR (300 MHz; $CDCl_3$) δ 1.21 (s, 3H), 1.44 (s, 9H), 1.79–1.87 (m, 1H), 2.04 (td, *J* = 5.0, 13.4 Hz, 1H), 2.50 (td, *J* = 4.9, 13.3 Hz, 1H), 2.59 (td, *J* = 5.1, 12.5 Hz, 1H), 3.63–3.76 (m, 2H), 4.30 (s, 1H), 4.65 (s, 1H), 5.33 (br s, 1H), 6.75 (dd, *J* = 2.1, 6.4 Hz, 2H), 7.02–7.04 (m, 2H).

**(2S)-2-amino-4-(4′-(3,4-dichlorobenzyloxy)phenyl)-2-methylbutan-1-ol (P053).** 3,4-Dichloro-benzyl bromide (20 mg, 81.3 μmol) was added dropwise to a suspension of (2S)-*t*-butyl(1-hydroxy-4-(4′-hydroxyphenyl)-2-methylbutan-2-yl) carbamate (24 mg, 81.3 μmol) and potassium carbonate (34 mg, 0.25 mmol) in dry DMF (0.8 mL). The suspension was stirred at room temperature for 15 h. The solution was diluted with water and extracted with ethyl acetate (×3). The organic extracts were combined and washed with water and brine, then dried ($Na_2SO_4$). The solvent was removed under reduced pressure and the crude material was purified by flash chromatography on silica gel, eluting with 30% ethyl acetate/*n*-hexane. After dissolving the purified material in methanol (0.5 mL), a solution of 2 M methanolic hydrochloric acid (0.12 mL, 0.24 mmol) was added dropwise to it and the resulting solution was heated at reflux for 4 h. After cooling to room temperature, the solvent was removed under reduced pressure. The residue was dissolved in chloroform and washed with saturated aqueous sodium bicarbonate solution (×2) and brine, then dried ($Na_2SO_4$). The solvent was removed under reduced pressure to afford the product as a white solid (16 mg, 57%). $[\alpha]_D^{25.5°C} = -2$ (0.5, $CHCl_3$); [1]H NMR (300 MHz; $CDCl_3$) δ 1.15 (s, 3H), 1.63–1.77 (m, 2H), 1.83 (br s, 3H), 2.60 (t, *J* = 8.5 Hz, 2H), 3.35 (d, *J* = 10.8 Hz, 1H), 3.41 (d, *J* = 10.8 Hz,

1H), 4.98 (s, 2H), 6.85–6.87 (m, 2H), 7.10–7.13 (m, 2H), 7.24–7.35 (m, 1H), 7.44 (d, $J$ = 8.3 Hz, 1H), 7.53 (d, $J$ = 1.6 Hz, 1H); $^{13}$C NMR (75 MHz; CDCl$_3$) δ 24.4, 29.5, 53.1, 68.8, 70.2, 77.4, 115.0, 126.7, 129.3, 129.5, 130.7, 135.2, 135.51, 135.54, 137.7, 156.6; IR (NaCl, neat) 3425 cm$^{-1}$; HRMS (ESI-MS): $m/z$ calcd for C$_{18}$H$_{22}$NO$_2$Cl$_2$ [M + H]$^+$ 354.1027, found 354.1017.

FTY720 and FB1 were purchased from Cayman Chemical Company. For biochemistry and cell treatments, compounds were prepared as stock solutions at 10 mM in DMSO, then diluted in water or cell culture medium.

**Ceramide synthase assays.** Ceramide synthase assays were carried out using an HPLC-based assay with a fluorescent NBD-dihydrosphingosine substrate[62]. All cell lines used for these assays were originally sourced from ATCC. The enzyme source was extracts of HEK293 cells exogenously expressing human CerS1, 2, and 6; U251 cells expressing human CerS4 (which did not express well in HEK293 cells); and COS cells expressing murine CerS1, 2, and 5. Human CerS expression plasmids (pCMV6 plasmids from Origene) were transfected using FuGene-6 transfection reagent, and cells were harvested 24 h after transfection in 20 mM Hepes pH 7.4, 10 mM KCl, 1 mM dithiothreitol, 3 mM β-glycerophosphate, and EDTA-free complete protease inhibitor cocktail (Roche). Cells were sonicated in a Diagenode Bioruptor sonicating bath for 5 min, after which cell extracts were cleared by centrifugation at 800 × $g$ for 10 min. Protein concentration in the supernatant was assayed using the bicinchoninic acid (BCA) assay, and the supernatant stored at −80 °C. COS cells were infected with lentiviral constructs expressing murine CerS1, 2, or 5, then homogenised as described above.

The reaction buffer consisted of 20 mM Hepes pH 7.4, 25 mM KCl, 2 mM MgCl$_2$, 0.5 mM dithiothreitol, 0.1% fatty acid free BSA, 10 μM NBD-dihydrosphingosine and 50 μM fatty acid free BSA. C18:0-CoA was used as the fatty acyl substrate for CerS1 and CerS4; C24:1-CoA for CerS2; and C16:0-CoA for CerS5 and CerS6. All reactions were run in triplicate. Test inhibitors were added, after which reactions were started with the addition of 2.5–25 μg lysate protein (optimised depending on the particular CerS isoform being assayed). Assays were run for 30 min at 37 °C, and stopped with the addition of 200 μL methanol. The reaction mixture was transferred to a glass HPLC vial with 300 μL fused glass insert (Thermo Fisher Scientific) and NBD-dihydroceramide reaction products were analysed on a Thermo Scientific Surveyor HPLC connected to a Shimadzu RF-10AXL fluorescent detector (set to gain of 3). A 3 × 150 mm Agilent XDB-C8 column (5 μM pore size) was used. Data was acquired over a 12 min chromatography run using: solvent A, 0.2% formic acid, 2 mM ammonium formate in water; solvent B, 0.2% formic acid, 1 mM ammonium formate in methanol. The gradient started at 20:80 A/B, increasing to 5:95 A/B over 2 min, then to 100% B over 6 min. The gradient was then held at 100% B for 2 min, before re-equilibration to 20:80 A/B for a further 2 min. Peak areas were integrated using Xcalibur software (Thermo Fisher Scientific).

**Sphingosine kinase assays.** Sphingosine kinase assays were performed using fluorescent NBD-dihydrosphingosine[63,64]. Reactions containing 50 mM Hepes pH 7.4, 0.01% BSA, 15 mM MgCl$_2$, 2 mM ATP, 10 μM NBD-dihydrosphingosine, and 10 μM FTY720 or P053, were started with addition of 10 ng recombinant human sphingosine kinase 1 or 2 (R&D Systems), and run for 30 min at 37 °C. Reactions were stopped with 100 μL methanol, and 5 μL amounts were resolved on Silica Gel 60 thin layer chromatography (TLC) plates using butanol:acetic acid:water (3:1:1). The fluorescent NBD-dihydrosphingosine 1-phosphate band was quantified using a BioRad ChemiDoc imaging system with excitation by UV light.

**Preparation of cortical neurons and quantification of D7-labelled ceramides.** Mouse cortical neurons were prepared from E16 pups[65] and seeded at 5 × 10$^6$ cells/well in poly-D-lysine coated six-well plates. Two weeks after seeding, neurons were pre-treated for 2 h with P053, then incubated for a further 1 h with 1 μM deuterated dihydrosphingosine (D7-dhSph). Lipid extraction and quantification of D7-ceramide using liquid chromatography-tandem mass spectrometry (LC-MS/MS)[34] is described below.

**Administration of P053 to mice.** Male C57BL6/J mice were obtained from the Animal Resources Centre in Perth (WA, Australia). Mice were housed at 22 ± 1 °C with a controlled 12:12 h light-dark cycle and had ad libitum access to water and either chow (8% calories from fat) or HFD (45% calories from fat)[66] for 4–6 weeks. P053 (5 mg/kg) was administered daily by oral gavage, with control animals receiving vehicle (2% DMSO in drinking water). Treatment was commenced at the same time as mice were randomised to dietary groups. The experiments were approved by the UNSW animal care and ethics committee (ACEC 15/48B), and followed guidelines issued by the National Health and Medical Research Council of Australia.

**Lipid extraction.** Lipids were extracted from frozen tissue samples or cells using a two-phase procedure with methyl-*tert*-butyl ether (MTBE)/methanol/water (10:3:2.5, v/v/v)[67]. Frozen tissue samples (~20 mg) were homogenized in 0.2 mL methanol containing 0.01% butylated hydroxytoluene (BHT), using a Precellys 24 homogenizer and Cryolys cooling unit (Bertin Technologies) with CK14 (1.4-mm ceramic) beads. The homogenates were spiked with an internal standard mixture

comprising 5 nmole 19:0/19:0 phosphatidylcholine and 2 nmole each of 17:0/17:0 phosphatidylethanolamine, 17:0/17:0 phosphatidylserine, 17:0/17:0 d5-diacylglycerol, 17:0/17:0 phosphatidylglycerol, 14:0/14:0/14:0/14:0 cardiolipin, 18:1/12:0 SM, 18:1/17:0 ceramide, 18:1/12:0 glucosylceramide, 18:1/12:0 lactosylceramide, 18:1/12:0 sulfatide (all from Avanti Polar Lipids, USA), and 17:0/17:0/17:0 TAG (Cayman Chemical, USA), then transferred to 10 mL screw cap glass tubes. The homogenization tubes containing beads were then washed with another 0.3 mL methanol, and this wash was combined with the first methanol extract, after which 1.7 mL MTBE was added and the samples were sonicated in an ice-cold sonicating water bath (Thermoline Scientific, Australia) for 30 min. Phase separation was induced by addition of 417 μL of mass spectrometry-grade water followed by vortexing and centrifugation at 1000 × $g$ for 10 min. The upper organic phase was collected into 5 mL glass tubes. The lower phase was re-extracted by adding 1 mL MTBE, 300 μL methanol, and 250 μL water. The organic phases were combined in the 5 mL glass tubes and dried under vacuum in a Savant SC210 SpeedVac (Thermo Scientific), then reconstituted in 500 μL methanol and stored at −20 °C until analysis.

HEK293 cells (10$^6$ cells/treatment) were treated for 24 h with P053, then washed with PBS and scraped into 0.6 mL methanol on ice. Lipids were then extracted using MTBE/methanol/water as described above[67].

**P053 quantification.** Plasma samples (20 μL) were extracted with 200 μL methanol containing 20 pmoles AAL(S)[27] as the internal standard. The mixtures were vortexed, then sonicated in ice water using a Bioruptor (Diagenode) for 2 min at 30 s intervals. Extracts were centrifuged at 1400 rpm for 20 min, 4 °C, to clear debris. Supernatants were transferred to 5 mL glass tubes. The remaining insoluble material was re-extracted with 1 mL 80% methanol as above, and the supernatant combined with the first extract. The extracts were evaporated under vacuum (Savant SC210 SpeedVac, Thermo Scientific). Dried extracts were reconstituted in 200 μL of 80% methanol/0.2% formic acid and stored at −20 °C. On the day of analysis, the extracts were vortexed thoroughly and centrifuged at 300 × $g$ for 20 min before transferring 100 μL of the supernatant to glass HPLC vials. To extract P053 from tissues, MTBE/methanol/water extraction was performed as described above, with AAL(S) as the internal standard.

LC-MS/MS was performed on a TSQ Quantum Access triple quadrupole mass spectrometer in positive ion mode, coupled to an Accela UPLC system (Thermo Scientific). Extracts (20 μL) were separated on a 2 × 100 mm Eclipse XDB-C8 column with 1.8 μm particle size (Agilent Technologies), using a 7.5 min binary HPLC gradient: 0 min, 70% B (20% A); 1.5 min, 70% B; 3.5 min, 95% B; 4 min, 95% B; 4.5 min, 70% B; 7.5 min 70% B. Mobile phase A: 2 mM ammonium formate/1% formic acid in water; mobile phase B: 1 mM ammonium formate/1% formic acid in methanol. The flow rate was 0.4 mL min$^{-1}$, increasing to 0.5 mL min$^{-1}$ from 5–7 min, and column oven 30 °C. P053 was quantified using an external standard curve covering the range 0.1–1000 nM, with all concentrations normalised to the 20 pmole AAL(S) internal standard. Visualisation and quantification of peaks was performed using Thermo Fisher's XCalibur software v2.2. Precursor/product ion $m/z$ values were as follows: 354.3 and 159.0 for P053; 294.0 and 161.1 for AAL(S); 380.3 and 255.2 for FTY720; 388.2 and 255.2 for FTY720-phosphate.

**Lipid quantification.** Lipidomic profiling was performed on a QExactive Plus mass spectrometer with heated electrospray ionization (HESI) probe and a Dionex UltiMate 2000 LC pump (Thermo Fisher Scientific). Extracts were resolved on a 2.1 × 100 mm Waters Acquity C18 UPLC column (1.7 μm pore size), using a binary gradient in which mobile phase A consisted of acetonitrile:water (60:40), 10 mM ammonium formate, 0.1% formic acid; and mobile phase B was isopropanol:acetonitrile (90:10), 10 mM ammonium formate, 0.1% formic acid[68]. The flow rate was 260 μL/min and column oven 55 °C. Data was acquired in full scan/data-dependent MS$^2$ mode (full scan resolution 70,000 FWHM, scan range 400–1200 $m/z$). The ten most abundant ions in each cycle were subjected to MS$^2$ using an isolation window of 1.4 $m/z$, collision energy 30 eV, resolution 17,500, maximum integration time 110 ms and dynamic exclusion window 10 s. An exclusion list of background ions was used based on a solvent blank. LipidSearch v4.1.30 software (ThermoFisher Scientific) was used for chromatogram alignment, peak identification and integration. Each lipid peak was manually verified for column elution time and characteristic fragment ions. Data were then exported to an Excel spreadsheet for normalisation of each lipid to its class-specific internal standard, and lipid concentrations were calculated relative to the internal standard.

Targeted quantification of ceramide, HexCer, and SM species in primary neurons, HEK293 cells, mouse brain, and gastrocnemius muscle lipid extracts was performed by LC-MS/MS on a TSQ Access triple quadrupole mass spectrometer (Thermo Fisher Scientific)[69]. Lipids were separated on a 3 × 150 mm Agilent XDB-C8 column (5 μM pore size) with a 10 min isocratic chromatography run using methanol containing 0.2% formic acid as the mobile phase. Precursor ions were the precise [M + H] species for each lipid. The $m/z$ 184.1 product ion was used for SM species, and the 264.3 $m/z$ product ion for ceramide and HexCer species. Deuterated (D7) ceramides were detected as the corresponding $m/z + 7$ ions.

Targeted quantification of sphingosine and dihydrosphingosine used the HPLC column and chromatography conditions as described above for P053. Precursor/

product ion $m/z$ values were: 300.3 and 264.3 for sphingosine; 302.3 and 266.3 for dihydrosphingosine.

**Body composition**. Body composition (lean mass and fat mass) was determined by EchoMRI-900 Body Composition Analyser (EchoMRI Corporation Pte Ltd, Singapore) in accordance with the manufacturer's instructions.

**Glucose metabolism and insulin action**. Mice were fasted for 6 h, then received 50 mg stable isotope-labeled glucose (6,6-$^2$H glucose, Sigma Aldrich) by oral gavage. Blood glucose was measured with a glucose meter (Accu-Check, Roche, NSW, Australia) prior to and at 15, 30, 45, 60 and 90 min following gavage. Additional blood (~10 μL) was collected from the tail tip prior to and at 15 and 60 min following glucose gavage. These samples were used to perform mass isotopologue analysis of plasma glucose via gas chromatography–mass spectrometry (GC–MS)[70]. Specifically, plasma glucose was separated into the labelled glucose load and the endogenous (liver/kidney-derived) unlabelled glucose to permit differentiation of glucose disposal from endogenous glucose production (Supplementary Figure 10). Fasting blood insulin was measured using an Ultrasensitive Mouse Insulin ELISA Kit (Crystal Chem, Illinois, USA).

**Hyperinsulinemic-euglycemic clamps**. At 5 weeks of diet/drug treatment, mice underwent surgery to implant catheters into the left carotid artery and right jugular vein. Catheters were flushed every 1–2 days with heparinised saline to maintain patency. Approximately 6–8 days post surgery, and after a ~5 h fast, a hyperinsulinemic-euglycemic clamp was conducted. Mice were conscious, unrestrained and were not handled during the procedure to minimize stress. At −90 min, a primed (5 μCi) continuous infusion (0.05 μCi/min) of [3,3H]-glucose (PerkinElmer) was commenced. Samples were collected at −30, −20, −10 and 0 min for basal glucose turnover (Rd) determination. At time 0, the rate of [3,3H]-glucose was increased (0.1 μCi/min) and mice received a primed (24 mU/kg)-continuous (6 mU/kg/min) infusion of insulin (Actrapid, Novo Nordisk, Copenhagen, Denmark) with euglycemia maintained at ~8 mM during the clamp through variable infusion of glucose (25% solution). Once blood glucose was stable, four sequential samples were taken to determine insulin-stimulated glucose turnover. A bolus of 2[$^{14}$C]deoxyglucose (10 μCi; PerkinElmer) was then administered and blood sampled at 2, 5, 10, 15, 20 and 30 min, prior to collection of muscles for determination of glucose uptake. Glucose uptake into muscle was determined via the following equation:

$$Rg' = \left(2\left[^{14}C\right]DGP_{tissue}/AUC\,2\left[^{14}C\right]DG_{plasma}\right) * [arterial\,glucose]$$

where 2[$^{14}$C]DGP$_{tissue}$ is the 2[$^{14}$C]DGP radioactivity in the muscle (in dpm/g), AUC 2[$^{14}$C]DG$_{plasma}$ is the area under the plasma 2[$^{14}$C]DG disappearance curve (in dpm/min/ml), and [arterial glucose] is the average blood glucose (in mM). Animal numbers for clamps were $n = 9$ Chow vehicle, $n = 8$ Chow P053, $n = 6$ HFD vehicle and $n = 6$ HFD P053.

**Palmitate oxidation and incorporation into TAG**. Tibialis muscles were homogenized in 19 volumes of 250 mM sucrose, 10 mM Tris-HCl and 1 mM EDTA, pH 7.4. 50 μL of tissue homogenate was incubated with 450 μL reaction buffer (111 mM Sucrose, 11.1 mM Tris-HCl, 5.56 mM KH$_2$PO$_4$, 1.11 mM MgCl$_2$, 88.9 mM KCl, 1.11 mM EDTA, 1.11 mM DTT, 2.22 mM ATP, 0.33% FA-free BSA, 2.22 mM L-carnitine, 0.056 mM CoA, 0.11 mM malate, 0.2 mM palmitate, 0.5 μCi mL$^{-1}$ [1-$^{14}$C]-palmitate, pH7.4) for 90 min at 30 °C, after which reactions were stopped with addition of 100 μL of 1 M perchloric acid. $^{14}$C-labelled CO$_2$ was collected in 100 μL of 1 M NaOH solution for 2 h. $^{14}$C-labelled CO$_2$, as well as $^{14}$C-labelled ASM present in the acidified supernatant, were quantified by scintillation counting.

To quantify palmitate oxidation in intact SkM, whole soleus muscles were dissected tendon to tendon and placed into a vial containing pre-warmed (30 °C) and pre-gassed (95% O$_2$, 5% CO$_2$) modified Krebs-Henseleit buffer (4% FA-free BSA, 5 mM glucose, 0.5 mM palmitate, pH 7.4). After a 30 min pre-incubation, muscles were transferred to vials containing 0.5 mCi/mL of [1-$^{14}$C]-palmitate in modified Krebs-Henseleit buffer for another 1 h. The reaction was stopped by removal of the muscles and addition of 1 M perchloric acid to the reaction mixture. $^{14}$C-labelled CO$_2$ was quantified as described above. The muscles were blotted dry and snap frozen in liquid nitrogen for subsequent lipid extraction and TLC. Muscles were homogenized using a Polytron (Kinematica, Littau-Lucerne, Switzerland) in 1.5 mL 2:1 chloroform-methanol (v/v). The homogenates were then sonicated in ice water for 15 min, after which 0.3 mL of distilled water was added, samples were vortexed, and centrifuged at $350 \times g$ for 10 min to induce phase separation. The upper aqueous phase was used to quantify ASM, and the lower organic phase was collected and dried under stream of N$_2$, then reconstituted in hexane and spotted on a Silica Gel 60 F$_{254}$ TLC plate. The TLC plate was developed in hexane/diethyl ether/glacial acetic acid (85:15:1), then air dried. TAG bands were visualised using iodine vapour staining, using an external standard to mark their position, then scraped into vials and quantified by liquid scintillation counting.

**Respiratory complex activities in permeabilized muscle fibres**. Permeabilized extensor digitorum longus (EDL) fibres were prepared and mitochondrial function was

analyzed according to published protocols with some modifications[71,72]. Briefly, EDL muscles were dissected tendon-to-tendon into ice-cold isolation buffer A (10 mM Ca-EGTA buffer, 20 mM imidazole, 20 mM taurine, 49 mM K-MES, 3 mM K$_2$HPO$_4$, 9.5 mM MgCl$_2$, 5.7 mM ATP, 15 mM phosphocreatine, 1 μM leupeptin, pH 7.1) and fibres were prepared immediately. Fibre bundles (~3 mg wet weight) were treated with saponin (50 μg/mL) for 20 min at 4 °C and subsequently washed in cold respiration medium B (0.5 mM EGTA, 3 mM MgCl$_2$.6H$_2$O, 20 mM taurine, 10 mM KH$_2$PO$_4$, 20 mM HEPES, 0.1% BSA, 15 mM potassium-lactobionate, 110 mM mannitol, 0.3 mM dithiothreitol, pH 7.1). Mitochondrial respiratory chain function was analyzed on a Clark-type electrode (Rank Brothers, UK) in situ, in respiration medium B at 37 °C, with the sequential addition of glutamate (10 mM), malate (5 mM), ADP (2 mM), rotenone (0.5 μM), succinate (10 mM), antimycin A (5 μM), N,N,N′,N′-tetramethyl-p-phenylenediamine dihydrochloride (0.5 mM TMPD), ascorbate (2 mM), and cytochrome c (10 μM). Fibres were recovered after polarography and results were expressed as nmoles of O$_2$/min/mg of tissue. Mitochondrial membrane integrity was verified by cytochrome c release test[71,72].

**Western blotting**. Tissues were homogenized in RIPA buffer (100 mM NaCl, 10 mM Tris, pH 7.4, 1% Triton X-100, 0.5% sodium deoxycholate, 0.1% SDS, 1 mM EDTA, 10% Glycerol) containing complete protease inhibitor cocktail (Roche) and phosphatase inhibitors (3 mM β-glycerophosphate, 1 mM sodium orthovanadate, 5 mM sodium fluoride) by bead-beating in a Precellys 24 at 4 °C. Extracts were centrifuged at 6500 rpm for 10 min at 4 °C. Supernatant was collected and protein concentrations determined with the Bicinchoninic acid (BCA) assay (Thermo Scientific). Protein lysates (15 μg) were resolved on SDS-PAGE, transferred to polyvinylidene difluoride (PVDF) membrane, and immunoblotted with antibodies against Complex I subunit NADH dehydrogenase (#ab110242, Abcam), Complex II succinate dehydrogenase subunit B (#ab14714, Abcam), Complex III subunit Core 2 Ubiquinol-cytochrome c reductase (#ab14745, Abcam), Complex IV Cytochrome c oxidase (#11967S, Cell Signaling Technology), Complex V ATP synthase α subunit (#ab14748, Abcam), PGC1α (#AB3242, Merck Millipore), AMPKα (#2532S, Cell Signaling Technology), phospho- AMPKα (Thr172) (#2535S, Cell Signaling Technology), ACC (#3662S, Cell Signaling Technology), phospho-ACC (Ser 79, which also detects Ser 212 in ACC2, the predominant ACC isoform in SkM) (#3661S, Cell Signaling Technology) and pan 14-3-3 (#sc-629, Santa Cruz Biotechnology). All antibodies were used at a dilution of 1:1000. A common loading control was included on every gel to account for variation in relative band intensities between different blots. Antibody-antigen binding was detected using ECL Reagent (GE Healthcare) and imaged on a Fujifilm Las-4000 CCD camera. Bands were quantified by densitometry with Fuji ImageQuantTL software. Full images of blots in Fig. 6 are shown in Supplementary Figure 12.

**Enzyme activity assays**. Liver and tibialis muscle were homogenized 1:19 (w/v) in 50 mM Tris-HCl, 1 mM EDTA, 0.1% Triton X-100, pH 7.4. The homogenates were subjected to three freeze-thaw cycles and centrifuged for 10 min at $7000 \times g$ at 4 °C. Supernatants were used to determine the activity of CS and βHAD[66]. CS activity of extracts was measured spectrophotometrically in 100 mM Tris–HCl, pH 8.2, 300 mM acetyl CoA, 1 mM MgCl$_2$, 1 mM EDTA, 0.1 mM 5,5′-dithio-bis (2-nitrobenzoic acid) (DTNB—ε 13.6 mM/cm at 412 nm) and 500 mM oxaloacetate, started by the addition of oxaloacetate. βHAD assays were conducted in 50 mM Imidazole buffer, pH 7.4, 1 mM EDTA, 0.15 mM NADH (ε 6.22 mM/cm at 340 nm) and 0.1 mM acetoacetyl-CoA, started by the addition of acetoacetyl-CoA.

**Real-time PCR**. PCR (40 cycles) was performed on a LightCycler 480 (Roche) using SensiFAST SYBR mastermix (Bioline). For quantification of CerS1 and CerS2 expression, total RNA was purified from mouse tissues with TRIzol reagent (Sigma-Aldrich), followed by DNase treatment (Promega), and reverse transcribed using random hexamer primers with FirstStrand cDNA synthesis kit (Roche), according to manufacturer's instructions. CerS1 and CerS2 gene expression was determined using a standard curve of Ct against input cDNA, and normalised to 18s RNA. For mitochondrial genes and transcriptional regulators in SkM, total RNA was prepared with TRIzol, and oligo dT primer was used for reverse transcription. Relative gene expression was determined using the ΔΔCt method, with 36B4 and Rpl13 as the reference genes. Samples with excessively high Ct values (Ct > 30 for the housekeeping genes) were removed from data analysis (two samples for Fig. 6a, four samples for Fig. 6g). Gene-specific primers were as follows:

| Target gene | Forward primer | Reverse primer |
| --- | --- | --- |
| 18s rRNA | GTAACCCGTTGAACCCCATT | CCATCCAATCGGTAGTAGCG |
| Cers1 | CCACCACACACATCTTTCGG | GGAGCAGGTAAGCGCAGTAG |
| Cers2 | ATGCTCCAGACCTTGTATGACT | CTGAGGCTTTGGCATAGACAC |
| 36B4 | GGCTCCAAGCAGATGCAGCAG | CCTGATAGCCTTGCGCATCATGG |
| Rpl13 | AGGAGGCGAAACAAGTCCAC | GGAGACTGGCAAAAGCCTTAAG |
| mt-Co2 | GCCGACTAAATCAAGCAACA | CAATGGGCATAAAGCTATGG |
| mt-Cytb | CATTTATTATCGCGGCCCTA | TGTTGGGTTGTTTGATCCTG |
| mt-Atp6 | GACGAACATGAACCCTAAT | TACGGCTCCAGCTCATAGT |
| mt-ND5 | ACCATGCTTATCCTCACTTCAG | AGTATTTGCGTCTGTTCGTCC |

**Table a** (continued)

| | | |
|---|---|---|
| Ndufb5 | TTTTCTCACGCGGAGCTTTC | ATAAAGAAGGCTTGACGACAAACA |
| Cox5b | GCTGCATCTGTGAAGAGGACAAC | CAGCTTGTAATGGGTTCCACAGT |
| Atp5o | AGGCCCTTTGCCAAGCTT | TTCTCCTTAGATGCAGCAGAGTACA |
| Cycs | GCAAGCATAAGACTGGACCAAA | TTGTTGGCATCTGTGTAAGAGAATC |
| Nfe2l2 | TAGATGACCATGAGTCGCTTGC | GCCAAACTTGCTCCATGTCC |
| Nrf1 | GTGCTGATGAAGACTCCCCTT | TGCCGTGGAGTTGAGGATGT |
| Gabpα1 | TGCCGTGGAGTTGAGGATGT | TCCTGCTCTTTTCTGTAGCCT |

**Mitochondrial DNA**. Total DNA was isolated using standard phenol:chloroform extraction, followed by RNase H (NEB) treatment. Relative levels of nuclear and mitochondrial DNA were determined by quantitative PCR on a LightCycler 480 (Roche) using SensiFAST SYBR mastermix (Bioline). Primers for quantification of genomic DNA were: 5′- GGCTGTATTCCCCTCCATCG-3′, and 5′- CCAGTTGG TAACAATGCCATGT-3′; and mitochondrial DNA: 5′- CCCAGCTACTACCATC ATTCAAGT-3′, and 5′- GATGGTTTGGGAGATTGGTTGATGT-3′[73]. Calibration curves of Ct against input DNA were prepared from a pooled sample of mouse SkM DNA, and used to calculate the relative abundance of mitochondrial to nuclear DNA in each sample.

**Statistical analyses**. HEK293 cell culture experiment: Levels of each lipid in P053-treated HEK293 cells were normalised to the mean of the vehicle control group, and each concentration was compared to the control using ANOVA with Dunnett's post-test. $P$ values were adjusted for multiple comparisons in GraphPad PRISM.

Lipidomic analysis of mouse tissue samples: Three comparisons were made for each lipid measured: chow control vs chow + P053, chow control vs HFD control, and HFD control vs HFD + P053. Data were log transformed, then subjected to two-tailed $t$-tests adjusted for multiple comparisons using GraphPad PRISM (Benjamini, Krieger and Yekutieli correction, $Q = 1\%$). A single outlier was identified using Grubbs test with high stringency ($P < 0.001$) and removed from the quadriceps sphingosine measurements.

Physiological and biochemical measures: Total ceramide, total sphingosine, and total TAG in SkM, western blotting and gene expression data, and physiological measures, were assessed by two-way ANOVA with diet as one variable and drug as the other. $P$ values for the main effect of diet (i.e., chow vs HFD) and drug (i.e., vehicle vs P053) are reported. Where post-tests were applied, Fisher's Least Significant Difference test was used.

Correlations: Spearman analysis was used to assess correlations between SkM ceramides and adipose tissue mass or % body fat.

**Data availability**. Lipidomic datasets for SkM and liver are provided in Supplementary Data File 2. Data used for bar graphs is provided in Supplementary Data File 1. Other raw data files supporting the results and conclusions presented in this manuscript are available from the corresponding authors upon reasonable request.

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

## Acknowledgements

This study was supported by National Health and Medical Research Council of Australia (NHMRC) project grant 1126135 (NT, KLH, JCM, CS-P, and ASD), a Diabetes Australia grant (NT, JCM, and ASD), ARC Future Fellowships (NT, CRB), a Prince of Wales Clinical School postgraduate scholarship (XYL), Australian Government APA postgraduate scholarships (HDT, SMB), an Australian Government Research Training Program postgraduate scholarship (ENT), an Alfred Deakin Postdoctoral Fellowship (GMK), and a NHMRC Early Career Fellowship (MKM). We gratefully acknowledge subsidised access to the UNSW Bioanalytical Mass Spectrometry Facility.

## Author contributions

Conceived and supervised the study: A.S.D., N.T., J.C.M. In vitro compound characterisation and lipidomics: X.Y.L., A.S.D., C-S.P. Cultured neurons: T.F. In vivo studies and tissue analysis: N.T., X.Y.L., B.O., A.E.B., C.E.F., H.G., J.T., H.P.M., T.A.C., A.D., G.M.K., C.R.B., K.L.H., G.J.C., M.K.M. Compound design and synthesis: H.D.T., E.N.T., S.M.B., J.C.M. Wrote the paper: A.S.D., N.T., X.Y.L., J.C.M.

## Additional information

**Competing interests:** The authors declare no competing interests.

