## [Peer Review File · Nature Communications]

Reviewers' comments:

Reviewer #1 (Remarks to the Author):

Turner et al examine the *in vitro* and *in vivo* effects of a new, selective inhibitor of Cers1, P503, on ceramide levels and glucose homeostasis. The manuscript presents convincing data indicating on the selectivity and potency of the compound in cellular models, as well as as an inhibitor of skeletal muscle C18-ceramide accumulation *in vivo*. Chronic administration of the compound increased mitochondrial beta-oxidation in muscle and slightly reduced adiposity. However, no effect was observed on glucose disposal, suggesting that C18-ceramides may not be important modulators of glucose homeostasis. A few concerns are noted

a. The relevant measure of muscle glucose disposal is insulin-sensitivity and uptake into muscle. The authors provide an insulin tolerance test, but the data are strangely presented (e.g. data at each time point aren't provided), and the error bars are huge. For example, the change between chow and HFD is not significant. If that measure isn't significant, it's unlikely that any drug effect could ever be detected. This is a major concern. The ideal solution would be a direct measure of glucose disposal using euglycemic clamps.

b. At times, the authors focus on the tissue driving the effect rather than the specific ceramides. For example, there are published data suggesting that subsarcolemmal C16-ceramides may drive the effect in muscle, and this is something that can't be ruled out by the experiments described herein. Rather than indicating that the data rule out muscle ceramides, the work should be more precise and indicate that it ruled out muscle C18 ceramides.

c. A broader analysis of the distribution of the drug effects in other tissues, including lipidomics and drug distribution, should be included. In particular, a largenumber of sphingolipid entities, particularly sphingomyelin, should be assessed in the mouse tissues. One would predict that changes in the acylation patterns of these more abundant sphingolipids could have important effects.

d. Surprisingly, the drug had no effects on C18-ceramides in other tissues. Presumably these are also made by the Cers1 in those locales, which is likely present, albeit at far lower concentrations. Why would the drug not influence them? Thesenegative findings should be addressed and discussed. It seems unlikely that the compounds would only distribute to SkM.

Minor Comments

1. The concentration units should be listed in Figure 1c.

Reviewer #2 (Remarks to the Author):

This is potentially an interesting paper. However, two areas need to be more carefully addressed, one concerning the lipidological aspects of the study and the other concerning mechanistic correlations.

1. While the data with P053 on selectivity of CerS inhibition is relatively convincing, the authors compare P053 with FTY720, which as the authors state, inhibits CerS via a mixed mode of inhibition, and with a relatively low potency. The authors have increased the potency of FTY720 on CerS1 by an order of magnitude using P053, but a much better comparison would be to compare with the potent CerS inhibitor, fumonosin B1. This is essential to determine the efficacy of CerS

inhibition.

2. Line 103. This assay does not measure ceramide levels but rather synthesis. Fix this all the way down to line 110. Also, in Fig. 2, it is not acceptable to show changes in ceramide synthesis as % change. This is particularly important since the absolute levels of synthesis of different chain length ceramides can differ by orders of magnitude. Thus, the authors must show absolute levels of ceramide synthesis, for each species in Figs. 2b-d. Moreover, why did the authors not measure total cellular ceramide levels after 24 h, rather than levels of synthesis using deuterated (D7) dihydrosphingosine - this should be a relatively trivial mass spec exp and would add a lot to the study, and should be carried out.

3. The last point above is particularly important since in the Lahiri paper, which the authors refer to, FTY720 causes elevation of ceramide levels in vivo (in cultured cells), measured by mass spec, and the authors suggest that this is due to the mode of inhibition of CerS activity. However, the current paper does not address this point in cultured cells.

4. The supplementary data on which Fig. 3 is based, does not appear to be complete in the PDF file which I received, and should be sent again. Irrespective of this, Figs 3c and d are unreadable. Fig. 3e is of great importance, and the decreases in SkM 18-ceramide seems clearcut, although there do appear to be changes in some other lipid species such as C24:1 and C24:0 - also the latter seems to change in liver and adipose tissue - why? Does this cast any doubt on the specificity of the inhibitor? The explanation that this may be due to elevated sphinganine levels is possible, which raises the question of why the authors never appear to have measured sphinganine (and sphingosine) levels - this is essential as it could profoundly change the interpretation of the results (i.e. which could be due to altered long chain base levels rather than C18-ceramides).

5. The changes in Figs. 4a-e are relatively small, and although they appear to be statistically significant, the authors need to explain their biological significance. Also, how do the authors mechanistically relate changes in C18-levels in SkM to the changes they see in Fig. 4? At the moment, these changes are nothing more than correlative and mechanism is lacking.

6. Does P053 interact with sphingosine kinase, i.e. does it inhibit SK activity or could it be phosphorylated by SK?

7. Again, the changes in Fig. 8a appear relatively small. Is it known whether this level of increased oxidation of 14C-palmitate can explain the 'the reduced lipid accretion in response to P053?' I.e. In other systems, what is the extent of change in palmitate oxidation required to induce such a biological response?

Minor points

1. Lines 47-50. Please use the correct references to indicate CerS specificity.

2. Line 58 - quote the original paper which showed this. Also, since palmitate is found at higher levels than ceramide in cells, it's difficult to imagine how fatty acid, rather than fatty acyl CoA availability, can affect ceramide synthesis

Reviewer #3 (Remarks to the Author):

The authors provide evidence for a first-in-class inhibitor for an isoform-specific ceramide synthase inhibitor which favors ceramide synthase 1 inhibition. They show its ability to dose-dependently inhibit C18 ceramide formation in neurons. When given in vivo c18 ceramides are decreased in

skeletal muscle. Given the higher expression of this isoform in muscle and brain they investigate the potential for this inhibitor to improve metabolism in muscle.

Only C22 ceramides are elevated in the muscle after the high fat diet treatment. As liver insulin resistance often precedes insulin resistance in muscle, one wonders if a longer duration of high fat diet would be needed to investigate the specific effects of cers1 inhibition on muscle insulin resistance.

It is not clear if muscle insulin resistance has developed. Figure 4g is referred to as "glucose disposal", however should be referred to as glucose tolerance even with the evaluation of the deuterated tracer. This is a poorly validated technique, appears to show an elevation of endogenous glucose at 20 minutes suggesting insulin resistance. Thus, it is still difficult to infer a muscle insulin resistance without additional supporting information.

The enhanced lipid oxidation is interesting, however the changes in mitochondrial complex abundance is not truly convincing. It would be nice to see functional assessment of mitochondrial oxidation by Seahorse or Oroboros. Additionally, Ming-hui Zou has implicated ceramide accumulation in other cell types to AMPK inhibition. Signaling components related to lipid oxidation and mitochondrial biogenesis are needed to provide a more comprehensive idea of what is going on- Pgc1a, acc, mitochondrial dna... In particular, I would be worried about this P53 acting as a PPARa agonist.

Cell autonomous effects of p53 on glucose uptake or insulin signaling in cultured myotubes would go a long way in making this muscle component of this work more convincing.

Response to reviewers comments

Reviewer #1 (Remarks to the Author):

Turner et al examine the *in vitro* and *in vivo* effects of a new, selective inhibitor of Cers1, P503, on ceramide levels and glucose homeostasis. The manuscript presents convincing data indicating on the selectivity and potency of the compound in cellular models, as well as as an inhibitor of skeletal muscle C18-ceramide accumulation *in vivo*. Chronic administration of the compound increased mitochondrial beta-oxidation in muscle and slightly reduced adiposity. However, no effect was observed on glucose disposal, suggesting that C18-ceramides may not be important modulators of glucose homeostasis. A few concerns are noted.

Comment: The relevant measure of muscle glucose disposal is insulin-sensitivity and uptake into muscle. The authors provide an insulin tolerance test, but the data are strangely presented (e.g. data at each time point aren't provided), and the error bars are huge. For example, the change between chow and HFD is not significant. If that measure isn't significant, it's unlikely that any drug effect could ever be detected. This is a major concern. The ideal solution would be a direct measure of glucose disposal using euglycemic clamps.

Response: While our initial data on glucose homeostasis/insulin action revealed no effect of P053 on fasting insulin, glucose tolerance or disposal of isotopically-labelled glucose during a glucose tolerance test, we agree with the reviewer that the optimal way to determine if P053 influences muscle insulin action is through hyperinsulinemic/euglycemic clamps. We have now performed clamps in chow- and fat-fed mice, with or without P053 treatment and demonstrate that P053 has no significant effect on glucose infusion rate, peripheral glucose disposal (Rd) or uptake of glucose into skeletal muscle (quadriceps, soleus) (Figure 4). Collectively, this data indicates that C18:0 ceramide in muscle is not a major regulator of whole body or muscle-specific insulin sensitivity.

Comment: At times, the authors focus on the tissue driving the effect rather than the specific ceramides. For example, there are published data suggesting that subsarcolemmal C16-ceramides may drive the effect in muscle, and this is something that can't be ruled out by the experiments described herein. Rather than indicating that the data rule out muscle ceramides, the work should be more precise and indicate that it ruled out muscle C18 ceramides.

Response: We agree with the reviewer that our data only rule out C18:0 ceramide as a mediator of muscle insulin resistance in high fat fed mice and have adjusted this throughout the revised manuscript.

Comment: A broader analysis of the distribution of the drug effects in other tissues, including lipidomics and drug distribution, should be included. In particular, a larger number of sphingolipid entities, particularly sphingomyelin, should be assessed in the mouse tissues. One would predict that changes in the acylation patterns of these more abundant sphingolipids could have important effects.

Response: We have added data on sphingomyelin for muscle, liver, adipose tissue (Fig. 3), and brain (Supplementary Fig. 7). Additional data for sphingolipid composition of a second skeletal muscle (gastrocnemius) has also been added in Supplementary Figure 4, using targeted LC-MS/MS analysis to confirm the results of untargeted lipidomic profiling that were already included in the manuscript. Hexosylceramide (HexCer) levels have also been shown for brain (Supplementary Fig.

7), and are available for liver in the Supplementary Data File. The C18 forms for SM were reduced with P053 treatment, but the overall reduction was less than for ceramide and was not statistically significant. The modest effect on SM levels was surprising to us, however we note that the reduction in C18 SM was also less than the change in ceramides in HEK293 cells treated with P053, suggesting either that the turnover rate for SM is slower than ceramide, or that steady state C18 SM levels are preferentially maintained at the expense of C18 ceramide. C18 is not a major form of HexCer, even in skeletal muscle, and the compound had no effect on HexCer. Additional discussion of these findings has been added on page 12.

With respect to compound distribution, we have now measured P053 concentration in a range of tissues, showing that the compound is higher in liver and kidney than in muscle and adipose tissue, and lowest in brain (Supplementary Figure 6). This is a typical drug distribution pattern and in agreement with the compound causing a reduction of C18 ceramide levels in muscle but not brain.

Comment: Surprisingly, the drug had no effects on C18-ceramides in other tissues. Presumably these are also made by the Cers1 in those locales, which is likely present, albeit at far lower concentrations. Why would the drug not influence them? These negative findings should be addressed and discussed. It seems unlikely that the compounds would only distribute to SkM.

Response: The reviewer raises a very interesting point. As noted above, we have data showing that P053 distributes in many tissues, but the reason that CerS1 inhibition does not reduce C18:0 ceramide in these other tissues is currently unclear. Whilst CerS1 catalyses the synthesis of C18 ceramide almost exclusively, other CerS isoforms are capable of catalysing C18 ceramide synthesis, as published in Mizutani et al, Biochem J, 2005, 390:263-267. In fact, ceramide synthases are believed to dimerise, and their activity towards different acyl-CoA substrates is regulated by this (Laviad et al, J Biol Chem, 2012, 287:21205-21033). For example Gosejacob et al (J Biol Chem 2016, 291: 6989-7003) showed that C18 ceramides in muscle and adipose tissue were reduced in CerS5 deficient mice on a HFD. Given the almost undetectable level of CerS1 expression/activity in tissues beyond brain and skeletal muscle (our data in Figure 3 and Schiffman et al Int J Biochem Cell Biol 2013, 45: 1886-1894, Laviad et al J Biol Chem 2008, 283: 5677-5684), it is possible that tissues may derive their C18:0 ceramide from CerS isoforms other than CerS1, or from other tissues, via uptake from the circulation (Boon et al Diabetes 2015, 62: 401-410). While this clearly requires further investigation, we have taken the reviewer's point on board and included additional discussion on page 12.

Minor Comments: The concentration units should be listed in Figure 1c.

Response: This has now been corrected in revised Table 1.

Reviewer #2 (Remarks to the Author):

This is potentially an interesting paper. However, two areas need to be more carefully addressed, one concerning the lipidological aspects of the study and the other concerning mechanistic correlations.

1. While the data with P053 on selectivity of CerS inhibition is relatively convincing, the

authors compare P053 with FTY720, which as the authors state, inhibits CerS via a mixed mode of inhibition, and with a relatively low potency. The authors have increased the potency of FTY720 on CerS1 by an order of magnitude using P053, but a much better comparison would be to compare with the potent CerS inhibitor, fumonosin B1. This is essential to determine the efficacy of CerS inhibition.

Response: This data is now presented in Table 1. The IC₅₀ for P053 on human CerS1 (540 nM) was similar to that of FB1 (220 nM). Whilst FB1 inhibited all CerS isoforms in the range 100 – 1000 nM, P053 was an order of magnitude less potent on the other isoforms.

2. Line 103. This assay does not measure ceramide levels but rather synthesis. Fix this all the way down to line 110. Moreover, why did the authors not measure total cellular ceramide levels after 24 h, rather than levels of synthesis using deuterated (D7) dihydrosphingosine - this should be a relatively trivial mass spec exp and would add a lot to the study, and should be carried out.

Response: This appears to be a misunderstanding. Fig. 2a shows results for ceramide synthesis from deuterated dihydrosphingosine, but 2b-d show endogenous ceramide levels in HEK293 cells. We have modified the legend for Fig. 2 to clearly indicate that results for endogenous ceramide, HexCer, and SM are presented in parts b – d.

Also, in Fig. 2, it is not acceptable to show changes in ceramide synthesis as % change. This is particularly important since the absolute levels of synthesis of different chain length ceramides can differ by orders of magnitude. Thus, the authors must show absolute levels of ceramide synthesis, for each species in Figs. 2b-d.

Response: We have added a new Supplementary Fig. 1 showing the endogenous ceramide, HexCer, and SM levels as pmoles/10⁵ cells. For Fig. 2b-d the levels were normalised to control because it is much easier in this format to gauge the relative effect of P053 on levels of different ceramide, SM, and HexCer species, i.e. much easier to present the specificity visually. This also allowed us to present combined data from two independent experiments in a single set of graphs. C18:0 and C20:0 ceramide are difficult to visualise on the same scale as other ceramides in cell lines, therefore requiring a broken axis (as in Supplementary Fig. 1) or separate graph. We can change Fig. 2b-d to absolute levels if the reviewers feel this is essential, but our preference is to present the absolute levels as a Supplementary Figure.

3. The last point above is particularly important since in the Lahiri paper, which the authors refer to, FTY720 causes elevation of ceramide levels in vivo (in cultured cells), measured by mass spec, and the authors suggest that this is due to the mode of inhibition of CerS activity. However, the current paper does not address this point in cultured cells.

Response: The definitive effect of P053 in both cultured cells and mouse skeletal muscle was a reduction in C18 ceramide. A modest increase in C24 ceramides was apparent in mouse skeletal muscle and cultured HEK293 cells. Across all of the in vitro enzyme assays, cell culture lipidomic assays, and in vivo lipidomic assays, our data clearly demonstrates that P053 is a selective inhibitor of CerS1. In the paper of Lahiri et al (J Biol Chem 2009, 284: 16090-16098), referred to above, it should be noted that FTY720 was used at a much higher dose of 25 micromolar for the cell culture assay.

4. The supplementary data on which Fig. 3 is based, does not appear to be complete in the

PDF file which I received, and should be sent again. Irrespective of this, Figs 3c and d are unreadable.

Response: We apologise that the supplementary data file did not convert properly and hope this is corrected in the revised version of the manuscript. The problem is in the uploading of the file. Fig. 3c and d have been enlarged in the revised version. The lipids that are significantly altered, as determined by t-tests adjusting for multiple comparisons, are labelled. We have also included extra lipidomic data (particularly SM) at the request of reviewer 1.

Fig. 3e is of great importance, and the decreases in SkM 18-ceramide seems clearcut, although there do appear to be changes in some other lipid species such as C24:1 and C24:0 - also the latter seems to change in liver and adipose tissue - why? Does this cast any doubt on the specificity of the inhibitor? The explanation that this may be due to elevated sphinganine levels is possible, which raises the question of why the authors never appear to have measured sphinganine (and sphingosine) levels - this is essential as it could profoundly change the interpretation of the results (i.e. which could be due to altered long chain base levels rather than C18-ceramides).

Response: We agree with the reviewer that measuring sphingosine and dihydrosphingosine (sphinganine) adds to the manuscript. We have now measured these lipids using targeted LC-MS/MS, showing that sphingosine, but not dihydrosphingosine, levels are modestly increased in skeletal muscle of mice administered P053 (Fig 3h). The increased C24 ceramide in muscle of mice fed a HFD plus P053 versus HFD plus vehicle would appear to be a direct consequence of increased substrate availability for CerS2 following CerS1 inhibition, as discussed on page 12. The fact that dihydrosphingosine and sphingosine levels were not substantially increased following CerS1 inhibition supports the notion that these substrates have been redirected for synthesis of other ceramides in muscle.

Although the increase in C24 ceramides in adipose tissue with P053 treatment was not statistically significant, we do agree that there appears to be an increase in the HFD group, not in the chow diet group. This non-significant trend is seen across most ceramide and SM species in the adipose tissue of P053-treated mice on a HFD. Note that the change is effectively a partial restoration of ceramide and SM levels to the “normal” levels as seen in mice on a chow diet, and may therefore be a secondary consequence of reduced adipose tissue mass in the HFD + P053 group compared to the HFD vehicle group.

In liver, there is no real evidence for increased C24 ceramide or SM, or any other species, with P053 treatment. By far the most significant change with P053 treatment, in terms of both P value and magnitude of change, is reduced C18 ceramide in muscle. This was established using both untargeted lipidomic profiling of multiple tissues, and targeted profiling of muscle and HEK293 cells, strongly supporting the specificity of P053 as an inhibitor of CerS1.

5. The changes in Figs. 4a-e are relatively small, and although they appear to be statistically significant, the authors need to explain their biological significance.

Response: As described in the results and discussion sections of our manuscript, the results in Fig. 4a-e show a significant inhibition of fat deposition in mice fed a high fat diet but treated with P053 (compared to mice fed a high fat diet and treated with vehicle control). The inhibition of fat deposition, along with reduced triglyceride levels in skeletal muscle, was observed reproducibly in

different mouse cohorts. To better illustrate the reduced fat gain with P053, we have included data on average fat pad weights on page 8/9 of the results. For both fat pads (inguinal and epididymal), the gain in mass on a HFD was reduced by a substantial 44% with P053 treatment.

Also, how do the authors mechanistically relate changes in C18-levels in SkM to the changes they see in Fig. 4? At the moment, these changes are nothing more than correlative and mechanism is lacking.

Response: As shown in Table 2, there are significant correlations between levels of specific ceramides in muscle, particularly C18 and C24, and the weight of the adipose tissues or overall % body fat. This indicates that muscle ceramide levels are strongly associated with adiposity, and the data is from mice that have not been treated with CerS1 inhibitor. The data that we subsequently present in Figure 4 demonstrate that this is likely a causal relationship, whereby reducing the C18 ceramide level in skeletal muscle with P053 reduces adiposity in mice fed a high fat diet. Our inhibitor affords the ability to selectively intervene in muscle ceramide content, so empowering us to test the causality of the association between muscle C18 ceramide content and adiposity. This is discussed more broadly in the context of other evidence on ceramides and fat mass in the second last paragraph of the discussion (p. 15).

We have also included new data strongly suggesting increased mitochondrial biogenesis in skeletal muscle of mice treated with P053, specifically, increased gene expression for respiratory complex subunits, increased oxygen consumption in permeabilised muscles, increased protein expression for Pgc-1alpha, a master regulator of mitochondrial biogenesis, increased gene expression for other transcriptional regulators of mitochondrial biogenesis (Gabp1alpha and Nfe2l2), and increased mitochondrial DNA content in HFD + P053 compared to HFD vehicle mice (all in Figure 6). So mechanistically, CerS1 inhibition boosts mitochondrial function and fatty acid oxidation in muscle. Please see directly below for further response to this question, as you raised a very closely-related point in comment 7.

7. Again, the changes in Fig. 8a appear relatively small. Is it known whether this level of increased oxidation of 14C-palmitate can explain 'the reduced lipid accretion in response to P053?' I.e. In other systems, what is the extent of change in palmitate oxidation required to induce such a biological response?

Response: We have now quantified the rate of incorporation of palmitate into the triglyceride pool (esterification; Fig. 5b), for direct comparison to the rate of oxidation in muscle (Fig. 5c). In absolute terms, the rate of esterification is very similar to the rate of oxidation, and that rate of oxidation was increased 23% with P053 treatment. In bioenergetic terms, muscle is a very large organ and accounts for ~20-30% of whole-body energy expenditure under standard conditions (Rolfe and Brown, Physiol Reviews 1997, 77: 731-758; ref. 53), so our results definitely point to a mechanism whereby, over the period of the study (4-6 weeks), the increase in mitochondrial capacity and enhanced rate of oxidation of dietary fatty acids in muscle is sufficient to significantly reduce their storage in adipose tissue or intramuscular triglycerides (Fig. 3j). To the best of our knowledge there are no prior studies that report comparable findings to those observed with P053 treatment. As noted in the discussion, other studies have manipulated fatty acid oxidation directly in muscle (e.g. through deletion of acetyl-CoA carboxylase or overexpression of CPT-1) without seeing any change in adiposity, but the difference is that these approaches have just induced a switch in fuel usage – i.e. increase in fatty acid usage at the expense of glucose. We have added in a

statement in the discussion (page 14) pointing out the importance of muscle to whole-body metabolism and how portioning of fatty acids towards oxidation over storage in muscle would be expected to have a significant impact on adiposity.

6. Does P053 interact with sphingosine kinase, i.e. does it inhibit SK activity or could it be phosphorylated by SK?

Response: We have now tested both of these possibilities, comparing P053 to FTY720. P053 shows no inhibitory activity towards SK1 or SK2 at 10 micromolar, whilst FTY720 at the same concentration shows greater than 50% inhibition of SK1 activity. This data is presented in Supplementary Figure 2. We were unable to detect the phosphate of P053 by mass spectrometry, either in cultured cells or mouse plasma. Using the same conditions to incubate HEK293 cells with FTY720 (2h), we detected a robust peak for FTY720-phosphate. This data is shown in Supplementary Figure 8.

Minor points

1. Lines 47-50. Please use the correct references to indicate CerS specificity.

Response: The references have been changed here.

2. Line 58 - quote the original paper which showed this. Also, since palmitate is found at higher levels than ceramide in cells, it's difficult to imagine how fatty acid, rather than fatty acyl CoA availability, can affect ceramide synthesis.

Response: Point taken, we have corrected the text and cited the original paper.

Reviewer #3 (Remarks to the Author):

The authors provide evidence for a first-in-class isoform-specific ceramide synthase inhibitor which favors ceramide synthase 1 inhibition. They show its ability to dose-dependently inhibit C18 ceramide formation in neurons. When given *in vivo* C18 ceramides are decreased in skeletal muscle. Given the higher expression of this isoform in muscle and brain they investigate the potential for this inhibitor to improve metabolism in muscle.

Only C22 ceramides are elevated in the muscle after the high fat diet treatment. As liver insulin resistance often precedes insulin resistance in muscle, one wonders if a longer duration of high fat diet would be needed to investigate the specific effects of cers1 inhibition on muscle insulin resistance. It is not clear if muscle insulin resistance has developed. Figure 4g is referred to as “glucose disposal”, however should be referred to as glucose tolerance even with the evaluation of the deuterated tracer. This is a poorly validated technique, appears to show an elevation of endogenous glucose at 20 minutes suggesting insulin resistance. Thus, it is still difficult to infer a muscle insulin resistance without additional supporting information.

Response: We (Turner et al Diabetologia, 2013, 56: 1638-1648) and others (Park et al Diabetes 2005, 54: 3530-3540; Lee et al Diabetes 2011, 60: 2474-2483) have shown that skeletal muscle becomes insulin resistant within ~3 weeks on a HFD and the current studies examined 4-6 weeks of high-fat feeding. As detailed in our response to Reviewer 1, we have now performed hyperinsulinemic/euglycemic clamp studies in mice fed a HFD for 6 weeks, demonstrating HFD-induced insulin resistance at the whole body and skeletal muscle level. Neither whole-body, nor

*skeletal muscle insulin resistance, was significantly affected by P053 treatment, despite the substantial reduction in C18 ceramide in muscle induced by this compound. We have also included additional lipidomic data for another muscle to show that after 6 weeks of treatment, C18 ceramide levels were significantly increased (Supplementary Figure 4). Our new data fully supports our initial conclusions that marked reductions in myocellular C18 ceramide have limited impact on muscle glucose homeostasis/insulin action in mice.*

**Comment: The enhanced lipid oxidation is interesting, however the changes in mitochondrial complex abundance is not truly convincing. It would be nice to see functional assessment of mitochondrial oxidation by seahorse or ouroboros. Additionally, Ming-hui Zou has implicated ceramide accumulation in other cell types to AMPK inhibition. Signaling components related to lipid oxidation and mitochondrial biogeneis are needed to provide a more comprenensive idea of what is going on- Pgc1a, acc, mitochondrial dna… In particular, I would be worried about this P053 acting as a PPARa agonist.**

*Response: As requested by the reviewer we have performed an array of additional measures of oxidative parameters. Using a Clark electrode system to assess mitochondrial oxygen consumption we have demonstrated enhanced ADP-stimulated (State 3) respiration with multiple substrates in permeabilised muscle preparations from P053 treated mice compared to vehicle-treated animals (Fig 5e). Coupled with our existing functional readout of oxidation of radiolabelled palmitate, these findings provide strong evidence for enhanced respiratory capacity in skeletal muscle in response to P053 treatment. As suggested, to further understand the basis for these changes, we have measured the mRNA expression of mitochondrial respiratory chain complexes, PGC-1alpha protein levels, mtDNA copy number and the expression of regulators of mitochondrial biogenesis (Figure 6). All of these parameters were upregulated in response to P053 treatment, suggesting that enhanced mitochondrial biogenesis is likely the underlying reason for the increased mitochondrial oxidative capacity induced by CerS1 inhibition. With respect to the query about whether these changes may be a consequence of activation of AMPK or PPAR alpha agonism, we have blotted for phosphorylation of AMPK and its downstream target ACC and find no influence of P053 on this pathway (Supplementary Fig 11). For PPAR alpha we have examined the protein expression of PMP70 (an abundant peroxisomal membrane protein) in liver lysates, as PPAR alpha is much more highly expressed in liver and we have clear evidence (Supplementary Fig 6) that P053 distributes into liver at levels that are higher that those present in skeletal muscle. As shown in the figure below, the expression of PMP70 was unaffected by P053 treatment, with liver samples from mice treated with a known PPAR alpha agonist WY14643 (WY) vs. vehicle (veh) controls shown on the right of the gel as positive controls. This data indicates that P053 does not activate PPARalpha.*

Comment: Cell autonomous effects of P053 on glucose uptake or insulin signaling in cultured myotubes would go a long way in making the muscle component of this work more convincing.

Response: We agree with the reviewer that experiments in muscle cells often allow additional mechanistic studies to be undertaken. We investigated this possibility in immortalised and primary murine myoblasts/myotubes during the course of the study and found that unlike skeletal muscle, where C18:0 ceramide is by the far the dominant ceramide species, cultured muscle cells contain much higher levels of other ceramides, particularly C16:0 (see figure below). This divergence in ceramide profile between cultured myotubes and muscle tissue in vivo has been reported by other groups (e.g. Hu et al. J Lipid Res, 2009, 50: 1852-1862; Mahfouz et al. PLoS ONE, 2014, 9: e101865; Park et al J Biol Chem 2016, 291: 23978-23988). Treatment with 1 μ M P053 did reduce C18:0 ceramide content in cultured primary myotubes by ~30% ($P = 0.02$), whilst having no significant effect on any other form of ceramide, including the dominant C16:0 form. However, as C18 ceramide is not the dominant form, and cultured muscle cells do not recapitulate the ceramide profile of skeletal muscle, CerS1 inhibition experiments with myotubes would have limited translatability to the in vivo setting.

REVIEWERS' COMMENTS:

Reviewer #1 (Remarks to the Author):

The authors have satisfied my concerns.

Reviewer #2 (Remarks to the Author):

I have carefully gone over the revision, and the authors have satisfactorily answered all my questions. They are to be congratulated on an excellent study

Reviewer #3 (Remarks to the Author):

The authors have satisfied all previous concerns